# Faster Discrete Convex Function Minimization with Predictions: The M-Convex Case

**Taihei Oki**
The University of Tokyo
Tokyo, Japan
oki@mist.i.u-tokyo.ac.jp

**Shinsaku Sakaue**
The University of Tokyo
Tokyo, Japan
sakaue@mist.i.u-tokyo.ac.jp

## Abstract

Recent years have seen a growing interest in accelerating optimization algorithms with machine-learned predictions. Sakaue and Oki (NeurIPS 2022) have developed a general framework that warm-starts the *L-convex function minimization* method with predictions, revealing the idea's usefulness for various discrete optimization problems. In this paper, we present a framework for using predictions to accelerate *M-convex function minimization*, thus complementing previous research and extending the range of discrete optimization algorithms that can benefit from predictions. Our framework is particularly effective for an important subclass called *laminar convex minimization*, which appears in many operations research applications. Our methods can improve time complexity bounds upon the best worst-case results by using predictions and even have potential to go beyond a lower-bound result.

## 1 Introduction

Recent research on *algorithms with predictions* [29] has demonstrated that we can improve algorithms' performance beyond the limitations of the worst-case analysis using predictions learned from past data. In particular, a surge of interest has been given to research on using predictions to improve the time complexity of algorithms, which we refer to as *warm-starts with predictions* for convenience. Since Dinitz et al. [11]'s seminal work on speeding up the Hungarian method for weighted bipartite matching with predictions, researchers have extended this idea to algorithms for various problems [7, 35, 10]. Sakaue and Oki [39] have found similarities between the idea and standard warm-starts in continuous convex optimization and extended it to *L-convex function minimization*, a broad class of discrete optimization problems studied in *discrete convex analysis* [31]. They thus have shown that warm-starts with predictions can improve the time complexity of algorithms for various discrete optimization problems, including weighted bipartite matching and weighted matroid intersection.

In this paper, we extend the idea of warm-starts with predictions to a new direction called *M-convex function minimization*, another important problem class studied in discrete convex analysis. The M-convexity is known to be in conjugate relation to the L-convexity. Hence, exploring the applicability of warm-starts with predictions to M-convex function minimization is crucial to broaden further the range of algorithms that can benefit from predictions, as is also mentioned in [39]. This paper mainly discusses an important subclass of M-convex function minimization called *laminar convex minimization* (Laminar), a large problem class widely studied in operations research (see references in Section 1.2). To make it easy to imagine, we describe the most basic form (Box) of Laminar,

$$(\text{Box}) \quad \underset{x \in \mathbb{Z}^n}{\text{minimize}} \ \sum_{i=1}^n f_i(x_i) \quad \text{subject to} \ \sum_{i=1}^n x_i = R, \ \ell_i \le x_i \le u_i \ (i = 1, \dots, n), \quad (1)$$

where $f_1, \dots, f_n : \mathbb{R} \to \mathbb{R}$ are univariate convex functions, $R \in \mathbb{Z}$, $\ell_1, \dots, \ell_n \in \mathbb{Z} \cup \{-\infty\}$, and $u_1, \dots, u_n \in \mathbb{Z} \cup \{+\infty\}$. Note that the variable $x \in \mathbb{Z}^n$ is an integer vector, which is needed when,

37th Conference on Neural Information Processing Systems (NeurIPS 2023).

Table 1: Our results and the best worst-case bounds for General, Laminar, Nested, and Box, where General refers to general M-convex function minimization discussed in Section 3.1. $n$ is the number of variables, $R$ specifies the equality constraint as in (1), and $m = |\{Y \in \mathcal{F} : |Y| \geq 2\}| = \mathrm{O}(n)$ is the number of additional constraints needed to convert Box into Nested and Laminar (see Section 4).

| PROBLEM | OUR RESULTS | WORST-CASE TIME COMPLEXITY |
|---|---|---|
| General | $\mathrm{O}(n\mathsf{SFM} + n^2\mathsf{EO}_f \cdot \|x^* - \hat{x}\|_1)$ | $\mathrm{O}\big(n^2 \log(L/n) \min\big\{n, \frac{n + \log^2(L/n)}{\log n}\big\}\mathsf{EO}_f\big)$ [43] |
| Laminar | $\mathrm{O}(n\|x^* - \hat{x}\|_1)$ | $\mathrm{O}\big(n^2 \log n \log \frac{mR}{n}\big)$ [18, 34][1] |
| Nested | $\mathrm{O}(n\|x^* - \hat{x}\|_1)$ | $\mathrm{O}(n \log m \log R)$ [46] |
| Box | $\mathrm{O}(n + \log n \cdot \|x^* - \hat{x}\|_1)$ | $\mathrm{O}\big(n \log \frac{R}{n}\big)$ [14, 19] |

for example, considering allocating $R$ indivisible resources to $n$ entities. As detailed later, adding some constraints and objectives to Box yields more general classes, Nested and Laminar, where the level of generality increases in this order. Streamlining repetitive solving of such problems by using predictions can provide substantial benefits of saving computation costs, as we often encounter those problems in, e.g., resource allocation [22], equilibrium analysis [16], and portfolio management [8].

## 1.1 Our contribution

We give a framework for accelerating M-convex minimization with predictions building on previous research [11, 39] (Section 3). We show that, given a vector $\hat{x} \in \mathbb{R}^n$ that predicts an optimal solution $x^* \in \mathbb{Z}^n$, the greedy algorithm (Algorithm 1) for M-convex function minimization finds an optimal solution in $\mathrm{O}(T_{\mathrm{init}} + T_{\mathrm{loc}}\|x^* - \hat{x}\|_1)$ time, where $T_{\mathrm{init}}$ and $T_{\mathrm{loc}}$ represent the time for converting $\hat{x}$ into an initial feasible solution and for finding a locally steepest descent direction, respectively. Since we can minimize $\|x^* - \hat{x}\|_1$ provably and approximately given optimal solutions to past instances [11, 23], this framework can provide better time complexity bounds benefiting from predictions. We also discuss how to apply our framework to general M-convex function minimization in Section 3.1.

Section 4 presents our main technical results. We apply our framework to Laminar, Nested, and Box and obtain time complexity bounds shown in Table 1. Our time complexity bounds can improve the existing worst-case bounds given accurate predictions. In particular, our $\mathrm{O}(n\|x^* - \hat{x}\|_1)$-time bound for Laminar improves the existing $\mathrm{O}(n^2 \log n \log \frac{mR}{n})$-time bound even if prediction error $\|x^* - \hat{x}\|_1$ is as large as $\mathrm{O}(n)$. Our result for Nested is a corollary of that for Laminar and improves the existing worst-case bound if $\|x^* - \hat{x}\|_1 = \mathrm{O}(1)$. In the case of Box, we can further reduce the time complexity to $\mathrm{O}(n + \log n \cdot \|x^* - \hat{x}\|_1)$ by modifying the algorithm for Laminar. Notably, our algorithm for Box runs in $\mathrm{O}(n)$ time if $\|x^* - \hat{x}\|_1 = \mathrm{O}(n/\log n)$, even though there exists an $\Omega(n \log \log(R/n^2))$-time lower bound for Box [19]. As far as we know, this is the first result that can go *beyond the lower bound* on the time complexity in the context of warm-starts with predictions. Experiments in Section 5 confirm that using predictions can improve empirical computation costs.

Few studies in the literature have made explicit improvements upon the theoretically fastest algorithms, even if predictions are accurate enough. The only exception is [7], whose shortest-path algorithm with predictions can improve the best worst-case time complexity by a couple of log factors. By contrast, our methods with accurate predictions can improve the best worst-case bounds by $\mathrm{O}(n)$ (up to log factors) in the Laminar case and potentially go beyond the lower-bound result in the Box case. Thus, our work not only extends the class of problems that we can efficiently solve using predictions but also represents a crucial step toward revealing the full potential of warm-starts with predictions. In this paper, we do not discuss the worst-case time complexity of our algorithms since we can upper bound it by executing standard algorithms with worst-case guarantees in parallel, as discussed in [39].

## 1.2 Related work

Algorithms with predictions [29], improving algorithms' performance by using predictions learned from past data, is a promising subfield in *beyond the worst-case analysis of algorithms* [38]. While this idea initially gained attention to improve competitive ratios of online algorithms [36, 2, 28, 1],

---

[1]While the worst-case analysis in [18, 34] focuses on separable objective functions, we can extend it to more general Laminar in (2) by introducing additional variables at the slight cost of setting $m = \Theta(n)$.

recent years have seen a surge of interest in improving algorithms' running time [11, 7, 39, 35, 10]. A comprehensive list of papers on algorithms with predictions is available at [27]. The most relevant to our work is [39], in which predictions are used to accelerate L-/L$^\natural$-convex function minimization, a large problem class including weighted bipartite matching and weighted matroid intersection. On the other hand, warm-starts with predictions remain to be studied for M-convex function minimization,[2] another essential class that is in conjugate relation to L-convex function minimization in discrete convex analysis [31]. Although our basic idea for utilizing predictions is analogous to the previous approach [11, 39], our algorithmic techniques to obtain the time complexity bounds in Table 1 for the specific M-convex function minimization problems are entirely different (see Section 4).

M-convex function minimization includes many important nonlinear integer programming problems, including Laminar, Nested, and Box, which have been extensively studied in the context of resource allocation [22]. A survey of recent results is given in [41]. Table 1 summarizes the worst-case time complexity bounds relevant to ours. Besides, faster algorithms for those problems under additional assumptions have been studied. For example, Schoot Uiterkamp et al. [41] showed that, if an objective function is a sum of $f(x_i + b_i)$ $(i = 1, \ldots, n)$ for some identical convex function $f$ and $b_i \in \mathbb{Z}$, we can solve Box, Nested, and Laminar with existing algorithms that run in $O(n)$ [4, 21], $O(n \log m)$ [46], and $O(n^2)$ [30] time, respectively. Hochbaum [19] gave an $O(n \log n \log \frac{R}{n})$-time algorithm for Laminar with separable objective functions and no lower bound constraints.[3] Even in those special cases, our results in Table 1 are comparable or better given that prediction errors $\|x^* - \hat{x}\|_1$ are small enough. There also exist empirically fast algorithms [42, 47], whose time complexity bounds are generally worse than the best results. Other problem classes with network and submodular constraints have also been studied [20, 30]. Extending our framework to those classes is left for future work.

Resource allocation with continuous variables has also been well-studied. One may think we can immediately obtain faster algorithms for discrete problems by accelerating continuous optimization algorithms for relaxed problems with predictions and using obtained solutions as warm-starts. However, this is not true since there generally exists an $O(n)$ gap in the $\ell_1$-norm between real and integer optimal solutions [30, Example 2.9], implying that we cannot always obtain faster algorithms for solving a discrete problem even if an optimal solution to its continuous relaxation is available for free.

## 2 Preliminaries

Let $N := \{1, \ldots, n\}$ be a finite ground set of size $n$. For $i \in N$, let $e_i$ be the $i$th standard vector, i.e., all zero but the $i$th entry set to one. For any $x \in \mathbb{R}^N$ and $X \subseteq N$, let $x(X) = \sum_{i \in X} x_i$. Let $\lfloor \cdot \rceil$ denote (element-wise) rounding to a closest integer. For a function $f : \mathbb{Z}^N \to \mathbb{R} \cup \{+\infty\}$ on the integer lattice $\mathbb{Z}^N$, its *effective domain* is defined as $\operatorname{dom} f := \{ x \in \mathbb{Z}^N : f(x) < +\infty \}$. A function $f$ is called *proper* if $\operatorname{dom} f \neq \emptyset$. For $Q \subseteq \mathbb{R}^N$, its *indicator function* $\delta_Q : \mathbb{R}^N \to \{0, +\infty\}$ is defined by $\delta_Q(x) := 0$ if $x \in Q$ and $+\infty$ otherwise.

### 2.1 M-convexity and greedy algorithm for M-convex function minimization

We briefly explain M-convex functions and sets; see [31, Sections 4 and 6] for details. We say a proper function $f : \mathbb{Z}^N \to \mathbb{R} \cup \{+\infty\}$ is *M-convex* if for every $x, y \in \operatorname{dom} f$ and $i \in \{ i' \in N : x_{i'} > y_{i'} \}$, there exists $j \in \{ j' \in N : x_{j'} < y_{j'} \}$ such that $f(x) + f(y) \geq f(x - e_i + e_j) + f(y + e_i - e_j)$. A non-empty set $Q \subseteq \mathbb{Z}^N$ is said to be *M-convex* if its indicator function $\delta_Q : \mathbb{Z}^N \to \{0, +\infty\}$ is M-convex. Conversely, if $f$ is an M-convex function, $\operatorname{dom} f$ is an M-convex set. An M-convex set always lies in a hyperplane defined by $\{ x \in \mathbb{R}^N : x(N) = R \}$ for some $R \in \mathbb{Z}$. It is worth mentioning that the M-convexity is built upon the well-known *basis exchange property* of matroids, and the base polytope of a matroid is the convex hull of an M-convex set.

The main subject of this paper is M-convex function minimization, $\min_{x \in \mathbb{Z}^N} f(x)$, where $f : \mathbb{Z}^N \to \mathbb{R} \cup \{+\infty\}$ is an M-convex function. Note that $\operatorname{dom} f \subseteq \mathbb{Z}^N$ represents the feasible region of the problem. For convenience of analysis, we assume the following basic condition.

**Assumption 2.1.** *An M-convex function* $f : \mathbb{Z}^N \to \mathbb{R} \cup \{+\infty\}$ *always has a unique minimizer* $x^*$.

---

[2]Similar to the L-/L$^\natural$-convex case, we can deal with $M^\natural$-*convex functions* by considering corresponding M-convex functions with one additional variable. See [31, Section 6.1] for details.

[3]An $O(n \log n)$-time algorithm for Laminar (and Nested) with quadratic objective functions was also proposed in [20], but later it turned out incorrect, as pointed out in [30].

This uniqueness assumption is common in previous research [39, 10] (and is also needed in [11, 7, 35], although not stated explicitly). In the case of Laminar, it is satisfied for generic, strictly convex objective functions. Even if not, there are natural tie-breaking rules, e.g., choosing the minimizer that attains the lexicographic minimum among all minimizers closest to the origin; we can implement this by adding $\epsilon\|x\|_2^2 + \sum_{i=1}^n \epsilon^{i+1}|x_i|$ for sufficiently small $\epsilon \in (0,1)$ to $f$, preserving its M-convexity. This is in contrast to the L-convex case, where arbitrarily many minimizers always exist (see [40]).

We can solve M-convex function minimization by a simple greedy algorithm shown in Algorithm 1, which iteratively finds a locally steepest direction, $-e_i + e_j$, and proceeds along it. If this update does not improve the objective value, the current solution is ensured to be the minimizer $x^* = \arg\min f$ due to the M-convexity [31, Theorem 6.26]. The number of iterations depends on the $\ell_1$-distance to $x^*$ as follows.

---
**Algorithm 1** Greedy algorithm
---
1: $x \leftarrow x^\circ$
2: **while** not converged **:**
3:     Find $i, j \in N$ that minimize $f(x - e_i + e_j)$
4:     **if** $f(x) \leq f(x - e_i + e_j)$ **:**
5:         **return** $x$
6:     $x \leftarrow x - e_i + e_j$

---

**Proposition 2.2** ([44, Corollary 4.2]). *Algorithm 1 terminates exactly in $\|x^* - x^\circ\|_1/2$ iterations.*

A similar iterative method is used in the L-convex case [39], whose number of iterations depends on the $\ell_\infty$-distance and a steepest direction is found by some combinatorial optimization algorithm (e.g., the Hopcroft–Karp algorithm in the bipartite-matching case). On the other hand, in the M-convex case, computing a steepest direction is typically cheap (as we only need to find two elements $i, j \in N$), while the number of iterations depends on the $\ell_1$-distance, which can occupy a larger fraction of the total time complexity than the $\ell_\infty$-distance. Hence, reducing the number of iterations with predictions can be more effective in the M-convex case. Section 3 presents a framework for this purpose.

Similar methods to Algorithm 1 are also studied in submodular function maximization [25]. Indeed, M-convex function minimization captures a non-trivial subclass of submodular function maximization that the greedy algorithm can solve (see [31, Note 6.21]), while it is NP-hard in general [32, 13]

## 3    Warm-start-with-prediction framework M-convex function minimization

We present a framework for warm-starting the greedy algorithm for M-convex function minimization with predictions. Although it resembles those of previous studies [11, 39], it is worth stating explicitly how the time complexity depends on prediction errors for M-convex function minimization.

We consider the following three phases as in previous studies: (i) obtaining an initial feasible solution $x^\circ \in \mathbb{Z}^N$ from a prediction $\hat{x} \in \mathbb{R}^N$, (ii) solving a new instance by warm-starting an algorithm with $x^\circ$, and (iii) learning predictions $\hat{x}$. The following theorem gives a time complexity bound for (i) and (ii), implying that we can quickly solve a new instance if a given prediction $\hat{x}$ is accurate.

**Theorem 3.1.** *Let $f : \mathbb{Z}^N \to \mathbb{R} \cup \{+\infty\}$ be an M-convex function that has a unique minimizer $x^* = \arg\min f$ and $\hat{x} \in \mathbb{R}^N$ be a possibly infeasible prediction. Suppose that Algorithm 1 starts from an initial feasible solution satisfying $x^\circ \in \arg\min\{ \|x - \lfloor\hat{x}\rceil\|_1 : x \in \mathrm{dom}\, f \}$. Then, Algorithm 1 terminates in $\mathrm{O}(\|x^* - \hat{x}\|_1)$ iterations. Thus, if we can compute $x^\circ$ in $T_{\mathrm{init}}$ time and find $i, j \in N$ that minimize $f(x - e_i + e_j)$ in Step 3 in $T_{\mathrm{loc}}$ time, the total time complexity is $\mathrm{O}(T_{\mathrm{init}} + T_{\mathrm{loc}}\|x^* - \hat{x}\|_1)$.*

*Proof.* Due to Proposition 2.2, the number of iterations is bounded by $\|x^* - x^\circ\|_1/2$. Thus, it suffices to prove $\|x^* - x^\circ\|_1 = \mathrm{O}(\|x^* - \hat{x}\|_1)$. By using the triangle inequality, we obtain $\|x^* - x^\circ\|_1 \leq \|x^* - \hat{x}\|_1 + \|\hat{x} - \lfloor\hat{x}\rceil\|_1 + \|\lfloor\hat{x}\rceil - x^\circ\|_1$. We below show that the right-hand side is $\mathrm{O}(\|x^* - \hat{x}\|_1)$. The second term is bounded as $\|\hat{x} - \lfloor\hat{x}\rceil\|_1 \leq \|\hat{x} - x^*\|_1$ since $x^* \in \mathbb{Z}^N$. As for the third term, we have $\|\lfloor\hat{x}\rceil - x^\circ\|_1 \leq \|\lfloor\hat{x}\rceil - x^*\|_1$ since $x^\circ \in \mathrm{dom}\, f$ is a feasible point closest to $\lfloor\hat{x}\rceil$ and $x^* \in \mathrm{dom}\, f$, and the right-hand side, $\|\lfloor\hat{x}\rceil - x^*\|_1$, is further bounded as $\|\lfloor\hat{x}\rceil - x^*\|_1 \leq \|\lfloor\hat{x}\rceil - \hat{x}\|_1 + \|\hat{x} - x^*\|_1 \leq 2\|\hat{x} - x^*\|_1$ due to the previous bound on the second term. Thus, $\|x^* - x^\circ\|_1 \leq 4\|\hat{x} - x^*\|_1$ holds. $\square$

Note that we round $\hat{x}$ to a closest integer point $\lfloor\hat{x}\rceil$ before projecting it onto $\mathrm{dom}\, f$, while rounding comes after projection in the L-/L$^\natural$-convex case [39]. This subtle difference is critical since rounding after projection may result in an infeasible integer point that is far from the minimizer $x^*$ by $\mathrm{O}(n)$ in the $\ell_1$-norm. To avoid this, we swap the order of the operations and modify the analysis accordingly.

Let us turn to phase (iii), learning predictions. This phase can be done in the same way as previous studies [11, 23]. In particular, we can learn predictions in an online fashion with the standard online subgradient descent method (OSD) as in [23], which works as follows in our case. Let $V \subseteq \mathbb{R}^N$ be a convex set that we expect to contain an optimal prediction. For any sequence of M-convex functions $f_1, \ldots, f_T$, we regard $L_t(\hat{x}) := \|x_t^* - \hat{x}\|_1$ for $t = 1, \ldots, T$ as loss functions, where $x_t^*$ is the minimizer of $f_t$. Fixing $\hat{x}_1 \in V$ arbitrarily, for $t = 1, \ldots, T$, OSD iteratively computes a subgradient $z_t \in \partial L_t(\hat{x}_t)$ and set $\hat{x}_{t+1} = \arg\min_{\hat{x} \in V} \|\hat{x}_t - \eta z_t - \hat{x}\|_2$, where $\eta > 0$ is the step size. OSD returns predictions $\hat{x}_1, \ldots, \hat{x}_T$ that enjoy a regret bound (see, e.g., [33]), and a sample complexity bound follows from online-to-batch conversion [5, 9]. Formally, the following proposition guarantees that we can provably learn predictions to decrease the time complexity bound in Theorem 3.1.

**Proposition 3.2** ([23]). *Let $f_t : \mathbb{Z}^N \to \mathbb{R} \cup \{+\infty\}$ for $t = 1, \ldots, T$ be a sequence of M-convex functions, each of which has a unique minimizer $x_t^* = \arg\min f_t$, and $V \subseteq \mathbb{R}^N$ be a closed convex set whose $\ell_2$-diameter is $D$. Then, OSD with $\eta = D/\sqrt{nT}$ returns $\hat{x}_1, \ldots, \hat{x}_T \in V$ that satisfy*

$$\sum_{t=1}^T \|x_t^* - \hat{x}_t\|_1 \leq \min_{\hat{x}^* \in V} \sum_{t=1}^T \|x_t^* - \hat{x}^*\|_1 + \mathrm{O}(D\sqrt{nT}).$$

*Furthermore, for any distribution $\mathcal{D}$ over M-convex functions $f : \mathbb{Z}^N \to \mathbb{R} \cup \{+\infty\}$, each of which has a unique minimizer $x_f^* = \arg\min f$, $\delta \in (0, 1]$, and $\varepsilon > 0$, given $T = \Omega\left(\left(\frac{D}{\varepsilon}\right)^2 n \log \frac{1}{\delta}\right)$ i.i.d. draws, $f_1, \ldots, f_T$, from $\mathcal{D}$, we can compute $\hat{x} \in V$ that satisfies*

$$\mathbb{E}_{f \sim \mathcal{D}} \|x_f^* - \hat{x}\|_1 \leq \min_{\hat{x}^* \in V} \mathbb{E}_{f \sim \mathcal{D}} \|x_f^* - \hat{x}^*\|_1 + \varepsilon$$

*with a probability of at least $1 - \delta$ via online-to-batch conversion (i.e., we apply OSD to $\|x_{f_t}^* - \cdot\|_1$ for $t = 1, \ldots, T$ and average the outputs).*

The convex set $V$ should be designed based on prior knowledge of upcoming instances. For example, previous studies [11, 39] set $V = [-C, +C]^N$ for some $C > 0$, which is expected to contain optimal solutions of all possible instances; then $D = 2C\sqrt{n}$ holds. In our case, we sometimes know that the total amount of resources is fixed, i.e., $x(N) = R$, and that every $x_i$ is always non-negative. Then, we may set $V = \{ x \in [0, R]^N : x(N) = R \}$, whose $\ell_2$-diameter is $D = R\sqrt{2}$.

## 3.1 Time complexity bound for general M-convex function minimization

We here discuss how to apply the above framework to general M-convex function minimization. The reader interested in the main results in Table 1 can skip this section and go to Section 4.

For an M-convex function $f : \mathbb{Z}^N \to \mathbb{R} \cup \{+\infty\}$, given access to $f$'s value and $\mathrm{dom}\, f$, we can implement the greedy algorithm with warm-starts so that both $T_{\mathrm{init}}$ and $T_{\mathrm{loc}}$ are polynomially bounded. Suppose that evaluating $f(x)$ for any $x \in \mathbb{Z}^N$ takes $\mathsf{EO}_f$ time. Then, we can find a steepest descent direction at any $x \in \mathrm{dom}\, f$ in $T_{\mathrm{loc}} = \mathrm{O}(n^2 \mathsf{EO}_f)$ time by computing $f(x - e_i + e_j)$ for all $i, j \in N$. As for the computation of $x^\circ$, we need additional information to identify $\mathrm{dom}\, f$ (otherwise, finding any feasible solution may require exponential time in the worst case). Since $\mathrm{dom}\, f$ is an M-convex set, we build on a fundamental fact that any M-convex set can be written as the set of integer points in the *base polyhedron* of an integer-valued *submodular function* [15, 31]. A set function $\rho : 2^N \to \mathbb{R} \cup \{+\infty\}$ is called *submodular* if $\rho(X) + \rho(Y) \geq \rho(X \cap Y) + \rho(X \cup Y)$ holds for $X, Y \subseteq N$, and its *base polyhedron* is defined as $\mathbf{B}(\rho) := \{ x \in \mathbb{R}^N : x(X) \leq \rho(X) \ (X \subseteq N), x(N) = \rho(N) \}$, where $\rho(\emptyset) = 0$ and $\rho(N) < +\infty$ are assumed. Thus, $\mathrm{dom}\, f$ is expressed as $\mathrm{dom}\, f = \mathbf{B}(\rho) \cap \mathbb{Z}^N$ with an integer-valued submodular function $\rho : 2^N \to \mathbb{Z} \cup \{+\infty\}$. We assume that, for any $x \in \mathbb{Z}^N$, we can minimize the submodular function $\rho + x$, defined by $(\rho + x)(X) := \rho(X) + x(X)$ for $X \subseteq N$, in SFM time. Then, we can obtain $x^\circ \in \mathrm{dom}\, f$ from $\lfloor \hat{x} \rfloor \in \mathbb{Z}^N$ in $T_{\mathrm{init}} = \mathrm{O}(n\mathsf{SFM})$ time, as detailed in Appendix A.1. Therefore, Theorem 3.1 implies the following bound on the total time complexity.

**Theorem 3.3.** *Given a prediction $\hat{x} \in \mathbb{R}^N$, we can minimize $f$ in $\mathrm{O}(n\mathsf{SFM} + n^2\mathsf{EO}_f \cdot \|x^* - \hat{x}\|_1)$ time.*

The current fastest M-convex function minimization algorithms run in $\mathrm{O}\left(n^3 \log \frac{L}{n} \mathsf{EO}_f\right)$ and $\mathrm{O}\left(\left(n^3 + n^2 \log \frac{L}{n}\right)\left(\log \frac{L}{n} / \log n\right)\mathsf{EO}_f\right)$ time [43], where $L = \max\{ \|x - y\|_\infty : x, y \in \mathrm{dom}\, f \}$. Thus, our algorithm runs faster if $\|x^* - \hat{x}\|_1 = \mathrm{o}(n)$ and $\mathsf{SFM} = \mathrm{o}(n^2 \mathsf{EO}_f)$. We discuss concrete situations where our approach is particularly effective in Appendix A.2.

# 4 Laminar convex minimization

This section presents how to obtain the time complexity bounds in Table 1 by applying our framework to laminar convex minimization (Laminar) and its subclasses, which are special cases of M-convex function minimization (see [31, Section 6.3]). We first introduce the problem setting of Laminar.

A *laminar* $\mathcal{F} \subseteq 2^N$ is a set family such that for any $X, Y \in \mathcal{F}$, either $X \subseteq Y$, $X \supseteq Y$, or $X \cap Y = \emptyset$ holds. For convenience, we suppose that $\mathcal{F}$ satisfies the following basic properties without loss of generality: $\emptyset \in \mathcal{F}$, $N \in \mathcal{F}$, and $\{i\} \in \mathcal{F}$ for every $i \in N$. Then, Laminar is formulated as follows:

$$\underset{x \in \mathbb{Z}^N}{\text{minimize}} \sum_{Y \in \mathcal{F}} f_Y(x(Y)) \quad \text{subject to} \quad x(N) = R, \ \ell_Y \le x(Y) \le u_Y \ (Y \in \mathcal{F} \setminus \{\emptyset, N\}), \quad (2)$$

where each $f_Y : \mathbb{R} \to \mathbb{R}$ ($Y \in \mathcal{F}$) is a univariate convex function that can be evaluated in $O(1)$ time, $R \in \mathbb{Z}$, and $\ell_Y \in \mathbb{Z} \cup \{-\infty\}$ and $u_Y \in \mathbb{Z} \cup \{+\infty\}$ for $Y \in \mathcal{F}$. We denote the objective function by $f_{\text{sum}}(x) := \sum_{Y \in \mathcal{F}} f_Y(x(Y))$. We let $f : \mathbb{Z}^N \to \mathbb{R} \cup \{+\infty\}$ be a function such that $f(x) = f_{\text{sum}}(x)$ if $x$ satisfies the constraints in (2) and $f(x) = +\infty$ otherwise; then, $f$ is M-convex and $\text{dom} f \subseteq \mathbb{Z}^N$ represents the feasible region of (2). Nested is a special case where $f_{\text{sum}}$ is written as $\sum_{i \in N} f_i(x_i)$ and $\{Y \in \mathcal{F} : |Y| \ge 2\} = \{Y_1, Y_2, \ldots, Y_m\}$ consists of nested subsets, i.e., $Y_1 \subset Y_2 \subset \cdots \subset Y_m$, and Box is a special case of Nested without nested-subset constraints. Note that our framework in Section 3 only requires the ground set $N$ to be identical over instances. Therefore, we can use it even when both objective functions and constraints change over instances.

It is well known that we can represent a laminar $\mathcal{F} \subseteq 2^N$ by a tree $T_{\mathcal{F}} = (\mathcal{V}, E)$. The vertex set is $\mathcal{V} = \mathcal{F} \setminus \{\emptyset\}$. For $Y \in \mathcal{V} \setminus \{N\}$, we call $X \in \mathcal{V}$ a *parent* of $Y$ if $X$ is the unique minimal set that properly contains $Y$; let $p(Y) \in \mathcal{V}$ denote the parent of $Y$. We call $Y \in \mathcal{V} \setminus \{N\}$ a *child* of $X$ if $p(Y) = X$. This parent–child relation defines the set of edges as $E = \{(X, Y) : X, Y \in \mathcal{V}, \ p(Y) = X\}$. Note that each $\{i\} \in \mathcal{V}$ corresponds to a leaf and that $N \in \mathcal{V}$ is the root. For simplicity, we below suppose the tree $T_{\mathcal{F}} = (\mathcal{V}, E)$ to be binary without loss of generality. If a parent has more than two children, we can recursively divide them into one and the others, which only doubles the number of vertices. Figure 1 illustrates a tree $T_{\mathcal{F}}$ of a laminar $\mathcal{F} = \{\emptyset, \{1\}, \{2\}, \{3\}, \{1, 2\}, \{1, 2, 3\}\}$.

Applications of Laminar include resource allocation [30], equilibrium analysis of network congestion games [16], and inventory and portfolio management [8]. We below describe a simple example so that the reader can better grasp the image of Laminar; we will also use it in the experiments in Section 5.

**Example: staff assignment.** We consider assigning $R$ staff members to $n$ tasks, which form the ground set $N$. Each task is associated with a higher-level task. For example, if staff members have completed tasks $1, 2 \in N$, they are assigned to a new task $Y = \{1, 2\}$, which may involve integrating the outputs of the individual tasks. The dependencies among all tasks, including higher-level ones, can be expressed by a laminar $\mathcal{F} \subseteq 2^N$. Each task $Y \in \mathcal{F}$ is supposed to be done by at least $\ell_Y (\ge 1)$ and at most $u_Y (\le R)$ members. An employer aims to assign staff members in an attempt to minimize the total perceived workload. For instance, if task $i \in N$ requires $c_i > 0$ amount of work and $x_i$ staff members are assigned to it, each of them may perceive a workload of $f_i(x_i) = c_i/x_i$. Similarly, the perceived workload of task $Y = \{1, 2\}$ is $f_Y(x(Y)) = c_Y/x(Y)$. The problem of assigning $R$ staff members to $n$ tasks to minimize the total perceived workload, summed over all tasks in $\mathcal{F}$, is formulated

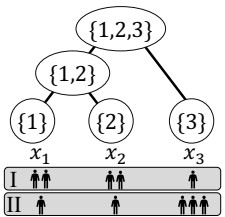

Figure 1: Image of tree $T_{\mathcal{F}}$. Each leaf $\{i\} \in \mathcal{V}$ ($i = 1, 2, 3$) has variable $x_i$. The lower part shows example assignments.

as in (2). Figure 1 illustrates two example assignments, I and II. Here, people assigned to $\{1\}$ and $\{2\}$ must do more tasks than those assigned to $\{3\}$, and hence assignment I naturally leads to a smaller total perceived workload than II. We can also use any convex function $f_Y$ on $[\ell_Y, u_Y]$ to model other objective functions. Making it faster to solve such problems with predictions enables us to manage massive allocations daily or more frequently.

Our main technical contribution is to obtain the following time complexity bound for Laminar via Theorem 3.1, which also applies to Nested since it is a special case of Laminar.

**Theorem 4.1.** *For Laminar, given a prediction $\hat{x} \in \mathbb{R}^N$, we can obtain an initial feasible solution $x^\circ \in \arg\min\{\|x - \lfloor \hat{x} \rfloor\|_1 : x \in \text{dom} f\}$ in $T_{\text{init}} = O(n)$ time and find a steepest descent direction in Step 3 of Algorithm 1 in $T_{\text{loc}} = O(n)$ time. Thus, we can solve Laminar in $O(n\|x^* - \hat{x}\|_1)$ time.*

We prove Theorem 4.1 by describing how to obtain an initial feasible solution and find a steepest descent direction in Sections 4.1 and 4.2, respectively. In Section 4.3, we further reduce the time complexity bound for Box. The algorithmic techniques we use below are not so complicated and can be implemented efficiently, suggesting the practicality of our warm-start-with-prediction framework.

## 4.1 Obtaining initial feasible solution via fast convex min-sum convolution

We show how to compute $x^\circ \in \mathrm{dom}\, f$ in $T_{\mathrm{init}} = \mathrm{O}(n)$ time. Given prediction $\hat{x} \in \mathbb{R}^N$, we first compute $\lfloor \hat{x} \rceil \in \mathbb{Z}^N$ in $\mathrm{O}(n)$ time and then solve the following special case of Laminar to obtain $x^\circ$:

$$\underset{x \in \mathbb{Z}^N}{\text{minimize}} \ \sum_{i \in N} |x_i - \lfloor \hat{x}_i \rceil| \quad \text{subject to} \ \ x(N) = R, \ \ell_Y \le x(Y) \le u_Y \ (Y \in \mathcal{F} \setminus \{\emptyset, N\}). \quad (3)$$

Note that it suffices to find an integer optimal solution to the continuous relaxation of (3) since all the input parameters are integers. Thus, we below discuss how to solve the continuous relaxation of (3).

Solving (3) naively may be as costly as solving the original Laminar instance. Fortunately, however, we can solve it much faster using the special structure of the $\ell_1$-norm objective function. The method we describe below is based on the fast convex min-sum convolution [45], which immediately provides an $\mathrm{O}(n \log^2 n)$-time algorithm for solving (3). We simplify it and eliminate the logarithmic factors by using the fact that the objective function has only two kinds of slopes, $\pm 1$.

We suppose that each non-leaf vertex $Y \in \mathcal{V}$ in $T_{\mathcal{F}} = (\mathcal{V}, E)$ has a variable $x_Y \in \mathbb{R}$, in addition to the original variables $x_i$ for leaves $\{i\} \in \mathcal{V}$. We consider assigning a univariate function $g : \mathbb{R} \to \mathbb{R} \cup \{+\infty\}$ of the following form to each vertex in $\mathcal{V}$:

$$g(x) = |x - b| + \delta_{[\ell, u]}(x), \quad (4)$$

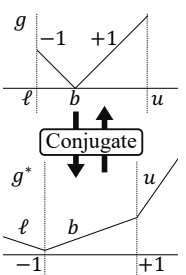

Figure 2: Conjugate relation of $g$ and $g^*$.

where $\ell, b, u \in \mathbb{Z} \cup \{\pm\infty\}$ and $\ell \le b \le u$. Note that if $g$ is given by (4) up to an additive constant, its convex conjugate $g^*(p) = \sup\{\langle p, x \rangle - g(x) : x \in \mathbb{R}\}$ is a piecewise-linear function whose slope is $\ell$ if $p \le -1$, $b$ if $-1 \le p \le +1$, and $u$ if $p \ge +1$ (where $\ell = b$ and/or $b = u$ can occur). Figure 2 illustrates this conjugate relation. We below construct such functions in a bottom-up manner on $T_{\mathcal{F}}$.

First, we assign function $g_i(x_i) = |x_i - \lfloor \hat{x}_i \rceil| + \delta_{[\ell_i, u_i]}(x_i)$ to each leaf $\{i\} \in \mathcal{V}$, which represents the $i$th term of the objective function and the constraint on $x_i$ in (3). Next, given two functions $g_X(x_X) = |x_X - b_X| + \delta_{[\ell_X, u_X]}(x_X)$ and $g_Y(x_Y) = |x_Y - b_Y| + \delta_{[\ell_Y, u_Y]}(x_Y)$ of $X, Y \in \mathcal{V}$ with an identical parent $X \cup Y \in \mathcal{V}$, we construct the parent's function as $g_{X \cup Y} = (g_X \square g_Y) + \delta_{[\ell_{X \cup Y}, u_{X \cup Y}]}$, where $(g_X \square g_Y)(x_{X \cup Y}) \coloneqq \min\{g_X(x_X) + g_Y(x_Y) : x_X + x_Y = x_{X \cup Y}\}$ is the infimal convolution. We can confirm that $g_{X \cup Y}$ also takes the form of (4) as follows. Since $g_X$ and $g_Y$ are of the form (4), $g_X^*$ and $g_Y^*$ have the same breakpoints, $\pm 1$ (see Figure 2). Furthermore, since $g_X \square g_Y = (g_X^* + g_Y^*)^*$ holds (e.g., [37, Theorem 16.4]), $g_X \square g_Y$ takes the form of (4) with $\ell = \ell_X + \ell_Y$, $b = b_X + b_Y$, and $u = u_X + u_Y$. Finally, adding $\delta_{[\ell_{X \cup Y}, u_{X \cup Y}]}$ preserves the form of (4). We can compute resulting $\ell$, $b$, and $u$ values of $g_{X \cup Y}$ in $\mathrm{O}(1)$ time, and hence we can obtain $g_Y$ for all $Y \in \mathcal{V}$ in a bottom-up manner in $\mathrm{O}(n)$ time. By construction, for each $Y \in \mathcal{V}$, $g_Y(x_Y)$ indicates the minimum objective value corresponding to the subtree, $(\mathcal{V}_Y, E_Y)$, rooted at $Y$ when $x_Y$ is given. That is, we have

$$g_Y(x_Y) = \min\Big\{ \textstyle\sum_{i \in Y} |x_i - \lfloor \hat{x}_i \rceil| \ : \ x(Y) = x_Y, \ \ell_{Y'} \le x(Y') \le u_{Y'} \ (Y' \in \mathcal{V}_Y \setminus \{Y\}) \Big\}$$

up to constants ignored when constructing $g_Y$, where $g_Y(x_Y) = +\infty$ if the feasible region is empty. Thus, $g_N(R)$ corresponds to the minimum value of (3), and our goal is to find integer values $x_Y$ for $Y \in \mathcal{V}$ that attain the minimum value when $x_N = R \in \mathbb{Z}$ is fixed.

Given $g_Y$ constructed as above, we can compute desired $x_Y$ values in a top-down manner as follows. Let $X \cup Y \in \mathcal{V}$ be a non-leaf vertex with two children $X$ and $Y$. Once $x_{X \cup Y} \in \mathrm{dom}\, g_{X \cup Y}$ is fixed, we can regard $\min\{g_X(x_X) + g_Y(x_Y) : x_X + x_Y = x_{X \cup Y}\}$ ($= g_{X \cup Y}(x_{X \cup Y})$) as univariate convex piecewise-linear minimization with variable $x_X \in \mathbb{R}$ (since $x_Y = x_{X \cup Y} - x_X$), which we can solve in $\mathrm{O}(1)$ time. Moreover, since $x_{X \cup Y}$ and all the parameters of $g_X$ and $g_Y$ are integers, we can find an integral minimizer $x_X \in \mathbb{Z}$ (and $x_Y = x_{X \cup Y} - x_X \in \mathbb{Z}$). Starting from $x_N = R \in \mathbb{Z}$, we thus compute $x_Y$ values for $Y \in \mathcal{V}$ in a top-down manner, which takes $\mathrm{O}(n)$ time. The resulting $x_i$ value for each leaf $\{i\} \in \mathcal{V}$ gives the $i$th element of a desired initial feasible solution $x^\circ \in \mathrm{dom}\, f$.

## 4.2 Finding steepest descent direction via dynamic programming

We present a dynamic programming (DP) algorithm to find a steepest descent direction in $T_{\mathrm{loc}} = \mathrm{O}(n)$ time. Our algorithm is an extension of that used in [30]. The original algorithm finds $i$ that minimizes $f(x - e_i + e_j)$ for a fixed $j$ in $\mathrm{O}(n)$ time. We below extend it to find a pair of $(i, j)$ in $\mathrm{O}(n)$ time.

Let $x \in \mathrm{dom}\, f$ be a current solution before executing Step 3 in Algorithm 1. We define a directed edge set, $A_x$, on the vertex set $\mathcal{V}$ as follows:

$$A_x = \{ (p(Y), Y) : Y \in \mathcal{V} \setminus \{N\}, \, x(Y) < u_Y \} \cup \{ (Y, p(Y)) : Y \in \mathcal{V} \setminus \{N\}, \, x(Y) > \ell_Y \}.$$

Note that $x - e_i + e_j$ is feasible if and only if $(\mathcal{V}, A_x)$ has a directed path from $\{i\} \in \mathcal{V}$ to $\{j\} \in \mathcal{V}$. We then assign an edge weight $w_{X,Y}$ to each $(X, Y) \in A_x$ defined as

$$w_{X,Y} = \begin{cases} f_Y(x(Y) + 1) - f_Y(x(Y)) & \text{if } X = p(Y), \\ f_X(x(X) - 1) - f_X(x(X)) & \text{if } Y = p(X). \end{cases}$$

By the convexity of $f_Y$, we have $w_{p(Y),Y} \geq w_{Y,p(Y)}$, i.e., there is no negative cycle. If $x - e_i + e_j$ is feasible, $f_{\mathrm{sum}}(x - e_i + e_j) - f_{\mathrm{sum}}(x)$ is equal to the length of a shortest path from $\{i\}$ to $\{j\}$ with respect to the edge weights $w_{X,Y}$ (see [30, Section 3.3]). Therefore, finding a steepest descent direction, $-e_i + e_j$, reduces to the problem of finding a shortest leaf-to-leaf path in this (bidirectional) tree $T_x := (\mathcal{V}, A_x)$. Constructing this tree takes $\mathrm{O}(n)$ time.

We present a DP algorithm for finding a shortest leaf-to-leaf path. For $Y \in \mathcal{V}$, we denote by $T_x(Y)$ the subtree of $T_x$ rooted at $Y$. Let $\mathsf{L}_\uparrow^Y$ be the length of a shortest path from a leaf to $Y$ in $T_x(Y)$, $\mathsf{L}_\downarrow^Y$ the length of a shortest path from $Y$ to a leaf in $T_x(Y)$, and $\mathsf{L}_\triangle^Y$ the length of a shortest path between any leaves in $T_x(Y)$. Clearly, $\mathsf{L}_\uparrow^Y = \mathsf{L}_\downarrow^Y = \mathsf{L}_\triangle^Y = 0$ holds if $Y$ is a leaf in $T_x$. For a non-leaf vertex $Y \in \mathcal{V}$, let $\mathcal{C}(Y)$ denote the set of children of $Y$ in $T_x$. We have the following recursive formulas:

$$\mathsf{L}_\uparrow^Y = \min_{\substack{X \in \mathcal{C}(Y): \\ (X,Y) \in A_x}} \{\mathsf{L}_\uparrow^X + w_{X,Y}\}, \quad \mathsf{L}_\downarrow^Y = \min_{\substack{X \in \mathcal{C}(Y): \\ (Y,X) \in A_x}} \{\mathsf{L}_\downarrow^X + w_{Y,X}\}, \quad \mathsf{L}_\triangle^Y = \min\Big\{\mathsf{L}_\uparrow^Y + \mathsf{L}_\downarrow^Y, \min_{X \in \mathcal{C}(Y)} \mathsf{L}_\triangle^X\Big\},$$

where we regard the minimum on an empty set as $+\infty$. Note that, if shortest leaf-to-$Y$ and $Y$-to-leaf paths in $T_x(Y)$ are not edge-disjoint, there must be a leaf-to-leaf simple path in $T_x(Y)$ whose length is no more than $\mathsf{L}_\uparrow^Y + \mathsf{L}_\downarrow^Y$ since no negative cycle exists. According to these recursive formulas, we can compute $\mathsf{L}_\uparrow^Y, \mathsf{L}_\downarrow^Y$, and $\mathsf{L}_\triangle^Y$ for all $Y \in \mathcal{V}$ in $\mathrm{O}(n)$ time by the bottom-up DP on $T_x$. Then, $\mathsf{L}_\triangle^N$ is the length of a desired shortest leaf-to-leaf path, and its leaves $\{i\}, \{j\} \in \mathcal{V}$ can be obtained by backtracking the DP table in $\mathrm{O}(n)$ time. Thus, we can find a desired direction $-e_i + e_j$ in $\mathrm{O}(n)$ time.

## 4.3 Faster steepest descent direction finding for box-constrained case

We focus on Box (1) and present a faster method to find a steepest descent direction, which takes only $T_{\mathrm{loc}} = \mathrm{O}(\log n)$ time after an $\mathrm{O}(n)$-time pre-processing. Note that we can obtain an initial feasible solution with the same method as in Section 4.1; hence $T_{\mathrm{init}} = \mathrm{O}(n)$ also holds in the Box case.

**Theorem 4.2.** *For Box, given a prediction $\hat{x} \in \mathbb{R}^N$, after an $\mathrm{O}(n)$-time pre-processing (that can be included in $T_{\mathrm{init}} = \mathrm{O}(n)$), we can find a steepest descent direction in Step 3 of Algorithm 1 in $T_{\mathrm{loc}} = \mathrm{O}(\log n)$ time. Thus, we can solve Box in $\mathrm{O}(n + \log n \cdot \|x^* - \hat{x}\|_1)$ time.*

*Proof.* In the Box case, $f(x - e_i + e_j) - f(x) = f_i(x_i - 1) - f_i(x_i) + f_j(x_j + 1) - f_j(x_j)$ holds if $x$ and $x - e_i + e_j$ are feasible. Furthermore, we only need to care about the box constraints, $\ell_i \leq x_i \leq u_i$ for $i = 1, \ldots, n$ (since $x(N) = R$ is always satisfied due to the update rule). Therefore, by keeping $\Delta_k^- := f_k(x_k - 1) - f_k(x_k) + \delta_{[\ell_k + 1, u_k]}(x_k)$ and $\Delta_k^+ := f_k(x_k + 1) - f_k(x_k) + \delta_{[\ell_k, u_k - 1]}(x_k)$ values for $k = 1, \ldots, n$ with two min-heaps, respectively, we can efficiently find $i \in \arg\min\{\Delta_k^-\}_{k=1}^n$ and $j \in \arg\min\{\Delta_k^+\}_{k=1}^n$; then, $-e_i + e_j$ is a steepest descent direction. More precisely, at the beginning of Algorithm 1, we construct the two heaps that maintain $\Delta_k^-$ and $\Delta_k^+$ values, respectively, and two arrays that keep track of the location of each element in the heaps; this pre-processing takes $\mathrm{O}(n)$ time. Then, in each iteration of Algorithm 1, we can find a steepest descent direction $-e_i + e_j$, update $\Delta_i^-, \Delta_i^+, \Delta_j^-$, and $\Delta_j^+$ values (by the so-called increase-/decrease-key operations), and update the heaps and arrays in $T_{\mathrm{loc}} = \mathrm{O}(\log n)$ time. Thus, Theorem 3.1 implies the time complexity. □

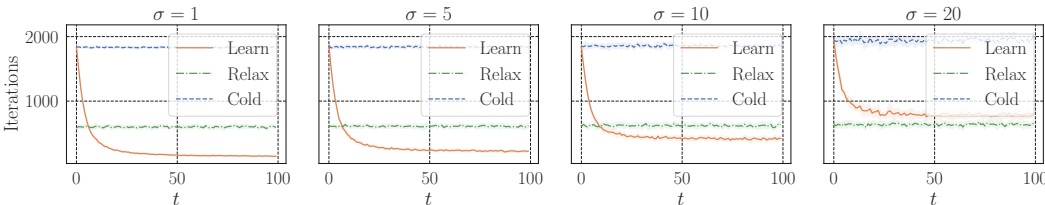

Figure 3: The number of iterations of the greedy algorithm initialized with Learn, Relax, and Cold. The curve and error band show the mean and standard deviation of 10 independent runs, respectively.

## 5  Experiments

We complement our theoretical results with experiments. We used MacBook Air with Apple M2 chip, 24 GB of memory, and macOS Ventura 13.2.1. We implemented algorithms in Python 3.9.12 with libraries such as NumPy 1.23.2. We used Gurobi 10.0.1 [17] for a baseline method explained later. The source code is available at https://github.com/ssakaue/alps-m-convex-code.

### 5.1  Staff assignment

We consider Laminar instances of the staff-assignment setting described in Section 4. Suppose that we have $R = 12800$ staff members and $n = 128$ tasks. Let $T_{\mathcal{F}} = (\mathcal{V}, E)$ be a complete binary tree with $n$ leaves. Define an objective function and inequality constraints as $f_{\mathrm{sum}}(x) = \sum_{Y \in \mathcal{V}} c_Y / x(Y)$ and $\ell_Y \le x(Y) \le R$ for $Y \in \mathcal{F} \setminus \{\emptyset, N\}$, respectively, with $c_Y$ and $\ell_Y$ values defined as follows. We set $c_Y = \max\{\sum_{i \in Y} i + \sigma u_a, 1\}$, where $u_a$ follows the standard normal distribution and $\sigma$ controls the noise strength. We let $\ell_Y = \min\{2^h + u_b, R/2^{n-h}\}$, where $h \in \{0, 1, \ldots, \log n\}$ is the height of $Y$ in $T_{\mathcal{F}}$ (a leaf $Y$ has $h = 0$) and $u_b$ is drawn uniformly at random from $\{0, 1, \ldots, 50\}$; the minimum with $R/2^{n-h}$ is taken to ensure that the feasible region is non-empty. We thus create a dataset of $T = 100$ random instances for each $\sigma \in \{1, 5, 10, 20\}$. We generate 10 such random datasets independently to calculate the mean and standard deviation of the results. The $T$ instances arrive one by one and we learn predictions from optimal solutions to past instances online. By design of $c_Y$, the $i$th entry of an optimal solution tends to be larger as $i$ increases, which is unknown in advance and should be reflected on predictions $\hat{x}$ by learning from optimal solutions to past instances.

We learn predictions $\hat{x}_t \in \mathbb{R}^N$ for $t = 1, \ldots, T$ by using OSD on $V = \{x \in [0, R]^N : x(N) = R\}$ with a step size of $0.01\sqrt{R/n}$, where the projection onto $V$ is implemented with a technique in [3]. We use the all-one vector multiplied by $R/n$ as an initial prediction, $\hat{x}_0 \in V$, and set the $t$th prediction, $\hat{x}_t$, to the average of past $t$ outputs, based on online-to-batch conversion. We denote this method by "Learn." We also use two baseline methods, "Cold" and "Relax", which obtain initial feasible solutions of the greedy algorithm as follows. Cold always uses $\hat{x}_0$ as an initial feasible solution. Relax is a variant of the continuous relaxation approach [30], the fastest method for Laminar with quadratic objectives. Given a new instance, Relax first solves its continuous relaxation (using Gurobi), where the objective function is replaced with its quadratic approximation at $\hat{x}_0$, and then converts the obtained solution into an initial feasible solution, as with our method. Note that Relax requires information on newly arrived instances, unlike Learn and Cold. Thus, Relax naturally produces good initial feasible solutions while incurring the overhead of solving new relaxed problems. We compare those initialization methods in terms of the number of iterations of the greedy algorithm.

Figure 3 compares Learn, Relax, and Cold for each noise strength $\sigma$. Learn always outperforms Cold, and it does even Relax if $\sigma \le 10$, suggesting that under moderate noise levels, learning predictions from past optimal solutions can accelerate the greedy algorithm more effectively than solving the relaxed problem of a new instance. The advantage of Learn decreases as $\sigma$ increases, as expected.

### 5.2  Resource allocation

We also present experiments using Nested instances of [47, Section 6.3], which include three types of problems, denoted by F, CRASH, and FUEL. The objective functions of F, CRASH, and FUEL are written as $\sum_{i=1}^n f_i(x_i)$ where $f_i(x_i) = x_i^4/4 + p_i x_i$, $f_i(x_i) = k_i + p_i/x_i$, and $f_i(x_i) = p_i \cdot c_i(c_i/x_i)^3$, respectively, with some input parameters $p_i$, $k_i$, $c_i$. F is a synthetic problem of optimizing the

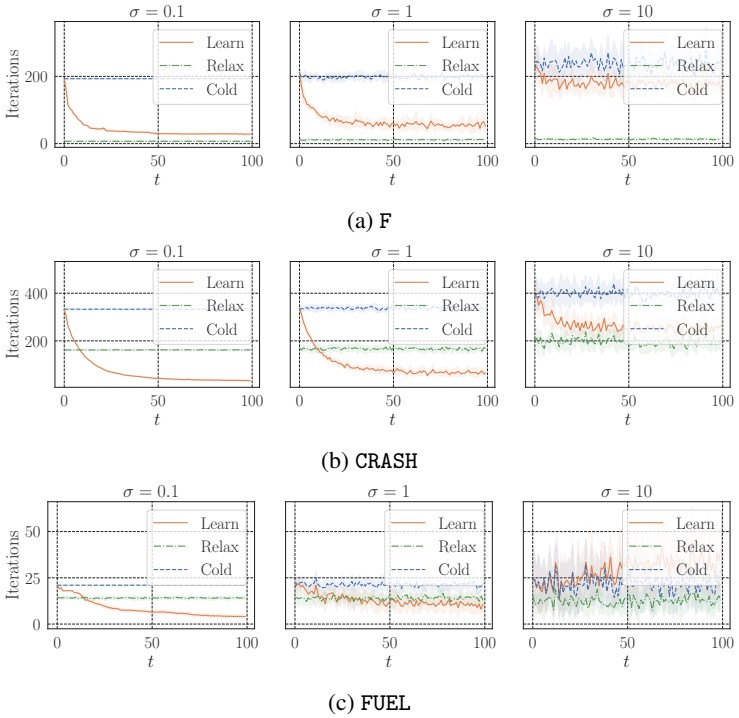

(a) F

(b) CRASH

(c) FUEL

Figure 4: The number of iterations of the greedy algorithm initialized with Learn, Relax, and Cold. The curve and error band show the mean and standard deviation of 10 independent runs, respectively.

fourth-order polynomial, while CRASH and FUEL come from real-world project crashing and ship speed optimization, as noted in [46]. We create Nested instances with $n = 100$ for F, CRASH, and FUEL based on the procedure of [47]. We then generate $T = 100$ instances by perturbing parameters defining constraints and objectives with Gaussian noises multiplied by $\sigma = 0.1, 1.0, 10.0$, which controls the noise strength. As with the previous experiments, we measure the number of iterations of the greedy algorithm initialized by Cold, Relax, and Learn over the 100 instances. Regarding Learn, we set OSD's step size in the same manner as the previous experiments, and each element of an initial prediction $\hat{x}_0$ is set to $\lfloor R/n \rfloor$ or $\lfloor R/n \rfloor + 1$ at random so that $\hat{x}_0(N) = R$ holds.

Figure 4 shows the results. Similar to the previous synthetic setting, Learn attains fewer iterations than Relax and Cold for CRASH and FUEL with moderate noise strengths ($\sigma = 0.1, 1.0$). As for F, Relax performs extremely well and surpasses Learn, probably because the synthetic fourth-order polynomial objective is easy to handle with the continuous-relaxation method used in Relax. Nevertheless, it is significant that Learn can surpass Relax for CRASH and FUEL, which come from real-world applications, under moderate noises.

## 6 Conclusion and limitations

We have extended the idea of warm-starts with predictions to M-convex function minimization. By combining our framework with algorithmic techniques, we have obtained specific time complexity bounds for Laminar, Nested, and Box. Those bounds can be better than the current best worst-case bounds given accurate predictions, which we can provably learn from past data. Experiments have confirmed that using predictions reduces the number of iterations of the greedy algorithm.

Since our focus is on improving theoretical guarantees with predictions, further study of practical aspects is left for future work. While we have used the standard OSD for learning predictions, we expect that more sophisticated learning methods can further improve the empirical performance. Also, extending the framework to other problem classes is an exciting future direction. A technical open problem is eliminating Assumption 2.1, although it hardly matters in practice. We expect the idea of [40] for the L-/L$^\natural$-convex case is helpful, but it seems more complicated in the M-convex case.

## Acknowledgments and Disclosure of Funding

The authors thank Satoru Iwata for telling us about the integrality of the dual problem of submodular function minimization. The authors also thank the anonymous reviewers for their helpful comments. This work was supported by JST ERATO Grant Number JPMJER1903 and JSPS KAKENHI Grant Number JP22K17853.

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
