# Faster Discrete Convex Function Minimization with Predictions: The M-Convex Case

**Taihei Oki**
The University of Tokyo
Tokyo, Japan
oki@mist.i.u-tokyo.ac.jp

**Shinsaku Sakaue**
The University of Tokyo
Tokyo, Japan
sakaue@mist.i.u-tokyo.ac.jp

## Abstract

Recent years have seen a growing interest in accelerating optimization algorithms with machine-learned predictions. Sakaue and Oki (NeurIPS 2022) have developed a general framework that warm-starts the *L-convex function minimization* method with predictions, revealing the idea's usefulness for various discrete optimization problems. In this paper, we present a framework for using predictions to accelerate *M-convex function minimization*, thus complementing previous research and extending the range of discrete optimization algorithms that can benefit from predictions. Our framework is particularly effective for an important subclass called *laminar convex minimization*, which appears in many operations research applications. Our methods can improve time complexity bounds upon the best worst-case results by using predictions and even have potential to go beyond a lower-bound result.

## 1 Introduction

Recent research on *algorithms with predictions* [29] has demonstrated that we can improve algorithms' performance beyond the limitations of the worst-case analysis using predictions learned from past data. In particular, a surge of interest has been given to research on using predictions to improve the time complexity of algorithms, which we refer to as *warm-starts with predictions* for convenience. Since Dinitz et al. [11]'s seminal work on speeding up the Hungarian method for weighted bipartite matching with predictions, researchers have extended this idea to algorithms for various problems [7, 35, 10]. Sakaue and Oki [39] have found similarities between the idea and standard warm-starts in continuous convex optimization and extended it to *L-convex function minimization*, a broad class of discrete optimization problems studied in *discrete convex analysis* [31]. They thus have shown that warm-starts with predictions can improve the time complexity of algorithms for various discrete optimization problems, including weighted bipartite matching and weighted matroid intersection.

In this paper, we extend the idea of warm-starts with predictions to a new direction called *M-convex function minimization*, another important problem class studied in discrete convex analysis. The M-convexity is known to be in conjugate relation to the L-convexity. Hence, exploring the applicability of warm-starts with predictions to M-convex function minimization is crucial to broaden further the range of algorithms that can benefit from predictions, as is also mentioned in [39]. This paper mainly discusses an important subclass of M-convex function minimization called *laminar convex minimization* (Laminar), a large problem class widely studied in operations research (see references in Section 1.2). To make it easy to imagine, we describe the most basic form (Box) of Laminar,

$$(\text{Box}) \quad \underset{x \in \mathbb{Z}^n}{\text{minimize}} \ \sum_{i=1}^n f_i(x_i) \quad \text{subject to} \ \sum_{i=1}^n x_i = R, \ \ell_i \le x_i \le u_i \ (i = 1, \ldots, n), \quad (1)$$

where $f_1, \ldots, f_n : \mathbb{R} \to \mathbb{R}$ are univariate convex functions, $R \in \mathbb{Z}$, $\ell_1, \ldots, \ell_n \in \mathbb{Z} \cup \{-\infty\}$, and $u_1, \ldots, u_n \in \mathbb{Z} \cup \{+\infty\}$. Note that the variable $x \in \mathbb{Z}^n$ is an integer vector, which is needed when,

37th Conference on Neural Information Processing Systems (NeurIPS 2023).

Table 1: Our results and the best worst-case bounds for General, Laminar, Nested, and Box, where General refers to general M-convex function minimization discussed in Section 3.1. $n$ is the number of variables, $R$ specifies the equality constraint as in (1), and $m = |\{Y \in \mathcal{F} : |Y| \geq 2\}| = O(n)$ is the number of additional constraints needed to convert Box into Nested and Laminar (see Section 4).

| PROBLEM | OUR RESULTS | WORST-CASE TIME COMPLEXITY |
|---------|-------------|----------------------------|
| General | $O(n\mathsf{SFM} + n^2\mathsf{EO}_f \cdot \|x^* - \hat{x}\|_1)$ | $O\left(n^2 \log(L/n) \min\left\{n, \frac{n+\log^2(L/n)}{\log n}\right\}\mathsf{EO}_f\right)$ [43] |
| Laminar | $O(n\|x^* - \hat{x}\|_1)$ | $O\left(n^2 \log n \log \frac{mR}{n}\right)$ [18, 34][1] |
| Nested | $O(n\|x^* - \hat{x}\|_1)$ | $O(n \log m \log R)$ [46] |

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

# A  Missing details in Section 3.1

## A.1  Projection onto base polyhedra via submodular function minimization

We prove Theorem 3.3 by presenting how to project the rounded point $\lfloor \hat{x} \rceil \in \mathbb{Z}^N$ of a prediction $\hat{x} \in \mathbb{R}^N$ onto the effective domain $\operatorname{dom} f$ of a general M-convex function $f : \mathbb{Z}^N \to \mathbb{R} \cup \{+\infty\}$. Recall that we have access to the submodular function $\rho : 2^N \to \mathbb{Z} \cup \{+\infty\}$ with $\operatorname{dom} f = \mathbf{B}(\rho) \cap \mathbb{Z}^N$ and that we can minimize $\rho + x$ in time $\mathsf{SFM}$ for any $x \in \mathbb{Z}^V$. Without loss of generality, we assume $\lfloor \hat{x} \rceil$ to be all zeros, denoted by $\mathbf{0}$; otherwise, we can replace $\rho$ with $\rho - \lfloor \hat{x} \rceil$ since $\mathbf{B}(\rho - \lfloor \hat{x} \rceil) = \{ x - \lfloor \hat{x} \rceil : x \in \mathbf{B}(\rho) \}$ holds (the *translation* of a base polyhedron [15]). We below discuss how to compute the $\ell_1$-projection, $x^\circ \in \arg\min\{ \|x\|_1 : x \in \mathbf{B}(\rho) \}$.

For $x \in \mathbf{B}(\rho)$, we have $\|x\|_1 = x(N) - 2x^-(N) = \rho(N) - 2x^-(N)$, where $x^- := (\min\{x_i, 0\})_{i \in N}$. Thus, it holds that $\arg\min\{ \|x\|_1 : x \in \mathbf{B}(\rho) \} = \arg\max\{ x^-(N) : x \in \mathbf{B}(\rho) \}$. The min-max theorem of submodular function minimization [12, 31, 15] claims that the minimum value of $\rho(X)$ over $X \subseteq N$ coincides with the maximum value of $x^-(N)$ over $x \in \mathbf{B}(\rho)$, and there exists an integral dual optimal solution if $\rho$ is integer-valued. Therefore, we can project $\lfloor \hat{x} \rceil = \mathbf{0}$ onto $\operatorname{dom} f$ by computing an integral optimal dual solution to submodular function minimization of $\rho$. However, no existing submodular function minimization algorithm directly returns an integral optimal dual solution, even if the objective function is integer-valued. Hence, we present a procedure to obtain an integral optimal dual solution that calls a submodular function minimization algorithm $\mathrm{O}(n)$ times.

We first rewrite the dual problem $\max\{ x^-(N) : x \in \mathbf{B}(\rho) \}$ as $\max\{ x(N) : x \in \mathbf{P}(\rho), x \le \mathbf{0} \}$, where $\le$ means the element-wise comparison and

$$\mathbf{P}(\rho) := \{ x \in \mathbb{R}^N : x(X) \le \rho(X) \ (X \subseteq N) \}$$

is the *submodular polyhedron* of $\rho$. Note that if $\tilde{x} \in \mathbf{P}(\rho)$ is an optimal solution to the rewritten problem, any point $x^\circ \in \mathbf{B}(\rho)$ with $x^\circ \ge \tilde{x}$ is an optimal solution to the original dual problem. The maximizer set of the rewritten problem is the base polyhedron of a submodular function $\rho^{\mathbf{0}}$ defined by $\rho^{\mathbf{0}}(X) := \min\{ \rho(Y) : Y \subseteq X \}$ for $X \subseteq N$ [15, Section 3.1]. Thus, we can reduce the evaluation of $\rho^{\mathbf{0}}(X)$ for $X \subseteq N$ to minimization of $\rho + x$ with $x \in \mathbb{Z}^N$ such that $x_i$ is $0$ for $i \in X$ and a sufficiently large constant for $i \in N \setminus X$. We can obtain an (extreme) point $\tilde{x} \in \mathbf{B}(\rho^{\mathbf{0}})$ with the *greedy algorithm* on the submodular polyhedron $\mathbf{P}(\rho^{\mathbf{0}})$; that is, we set $\tilde{x}_i = \rho^{\mathbf{0}}(\{1, \ldots, i\}) - \rho^{\mathbf{0}}(\{1, \ldots, i-1\})$ for $i \in N$ [15, Section 3.2]. Thus, we can compute $\tilde{x}$ in $\mathrm{O}(n\mathsf{SFM})$ time. We then convert $\tilde{x}$ back into an optimal solution to the original dual problem by computing a point $x^\circ \in \mathbf{B}(\rho)$ with $x^\circ \ge \tilde{x}$. To this end, we again use (another form of) the greedy algorithm: initializing $x^\circ$ as $\tilde{x}$, for $i = 1$ to $n$, we put $x^\circ \leftarrow x^\circ + \hat{c}(x^\circ, e_i)e_i$, where

$$\hat{c}(x^\circ, e_i) := \max\{ \lambda \in \mathbb{R} : x^\circ + \lambda e_i \in \mathbf{P}(\rho) \} = \min\{ \rho(X) - x^\circ(X) : X \subseteq N, i \in X \}$$

is the *saturation capacity* [15]. We can compute $\hat{c}(x^\circ, e_i)$ in time $\mathsf{SFM}$ in the same way as evaluation of $\rho^{\mathbf{0}}(X)$. Since $\tilde{x}$ and $x^\circ$ are integral, the resulting $x^\circ$ is the desired projection. To conclude, we can compute a projection via $\mathrm{O}(n)$ calls to submodular function minimization, i.e., $T_{\mathrm{init}} = \mathrm{O}(n\mathsf{SFM})$.

## A.2  Discussion on time complexity bounds for general M-convex function minimization

We discuss some scenarios where our algorithm given in Section 3.1 can be faster than general M-convex function minimization algorithms. For a general M-convex function $f : \mathbb{Z}^N \to \mathbb{R} \cup \{+\infty\}$, our algorithm takes $T_{\mathrm{init}} = \mathrm{O}(n\mathsf{SFM})$ time for projection and $T_{\mathrm{loc}} = \mathrm{O}(n^2 \mathsf{EO}_f)$ time for finding a steepest descent direction, which results in the total time complexity of $\mathrm{O}(T_{\mathrm{init}} + T_{\mathrm{loc}}\|x^* - \hat{x}\|_1) = \mathrm{O}(n\mathsf{SFM} + n^2 \mathsf{EO}_f \cdot \|x^* - \hat{x}\|_1)$ as described in Theorem 3.3. Here, for a given $x \in \mathbb{Z}^N$, $\mathsf{EO}_f$ and $\mathsf{SFM}$ denote the time to evaluate $f(x)$ and to minimize $\rho + x$, respectively, where $\rho : 2^N \to \mathbb{R} \cup \{+\infty\}$ is the submodular function representing $\operatorname{dom} f$. The current fastest M-convex function minimization algorithms run in $\mathrm{O}\!\left(n^3 \log \frac{L}{n} \mathsf{EO}_f\right)$ and $\mathrm{O}\!\left(\left(n^3 + n^2 \log \frac{L}{n}\right)\left(\log \frac{L}{n} / \log n\right) \mathsf{EO}_f\right)$ time [43],[4] where $L = \max\{ \|x - y\|_\infty : x, y \in \operatorname{dom} f \}$. Therefore, our algorithm runs faster if $\|x^* - \hat{x}\|_1 = \mathrm{o}(n)$ and $\mathsf{SFM} = \mathrm{o}(n^2 \mathsf{EO}_f)$ (or $T_{\mathrm{init}} = \mathrm{o}(n^3 \mathsf{EO}_f)$). We below list some situations where $T_{\mathrm{init}} = \mathrm{o}(n^3 \mathsf{EO}_f)$ or $\mathsf{SFM} = \mathrm{o}(n^2 \mathsf{EO}_f)$ can occur.

---

[4]The algorithms in [43] require a feasible initial point $x^\circ \in \operatorname{dom} f$ as input. If the finite- and integer-valued submodular function $\rho : 2^N \to \mathbb{Z}$ representing $\operatorname{dom} f$ is given instead of $x^\circ$, we can obtain a point in $\operatorname{dom} f$ by the greedy algorithm on $\mathbf{P}(\rho)$ that evaluate $\rho$'s value $\mathrm{O}(n)$ times [15].

First, consider the case where $\mathrm{dom}\, f$ is fixed over all instances. In this situation, we can compute $x^\circ \in \arg\min\{\, \|x - \lfloor \hat{x} \rceil\|_1 \, : \, x \in \mathrm{dom}\, f \,\}$ from a prediction $\hat{x}$ before a new actual instance of $f$ is revealed, which means that the projection can be included in the phase of computing a prediction $\hat{x}$. As a result, we can exclude the time for obtaining an initial solution from the time complexity bound of Theorem 3.1, i.e., $T_{\mathrm{init}} = 0$.

The second scenario is the case where we can represent an objective M-convex function $f$ as

$$f(x) = \begin{cases} h(x) & (x \in \mathbf{B}(\rho)), \\ +\infty & (\text{otherwise}) \end{cases} \tag{5}$$

using an M-convex function $h : \mathbb{Z}^N \to \mathbb{R}$ with $\mathrm{dom}\, h = \mathbb{Z}^N$ and a submodular function $\rho : 2^N \to \mathbb{Z} \cup \{+\infty\}$. Although the function $f$ in the form of (5) is not always M-convex (but $M_2$-convex), it is so in some special cases where, e.g., $h$ is separable convex and/or $\rho$ is modular (linear). Notably, the separable convex case is widely studied in resource allocation [20, 30, 41]. In this case, evaluating $f(x)$ for a given $x \in \mathbb{Z}^N$ involves the membership testing of $x$ for $\mathbf{B}(\rho)$, which costs SFM time since $x \in \mathbf{B}(\rho)$ is equivalent to $x(N) = \rho(N)$ and $\min_{X \subseteq N}(\rho - x)(X) \geq 0$. Thus, $\mathsf{SFM} \leq \mathsf{EO}_f$ holds, and hence we can assume $\mathsf{SFM} = o(n^2 \mathsf{EO}_f)$. We, however, remark that algorithms specialized for this case can run faster than the general M-convex function minimization algorithms (see, e.g., [41, Section 4.5]), and hence ours is not necessarily the best choice. We omit detailed comparisons with them since they involve more case-specific discussions.

The last scenario is the case where $\mathsf{EO}_f$ is sufficiently larger than the time to evaluate $\rho(X)$ for a given $X \subseteq N$, denoted by $\mathsf{EO}_\rho$. The fastest submodular function minimization algorithm runs in $\mathsf{SFM} = \mathrm{O}\!\left(n^3 \log^2 n \cdot \mathsf{EO}_\rho + n^4 \log^{\mathrm{O}(1)} n\right)$ time [26]. Therefore, we have $\mathsf{SFM} = o(n^2 \mathsf{EO}_f)$ if $\mathsf{EO}_f$ is asymptotically larger than $n \log^2 n \cdot \mathsf{EO}_\rho + n^2 \log^{\mathrm{O}(1)} n$. More efficient submodular function minimization algorithms are available if $\rho$ enjoys some special structures; for example, $\rho$ is the rank function of certain matroids. There also exists an empirically fast algorithm for submodular function minimization [6, 24], although its time complexity is worse than that of [26].