# OpenReview forum: "Faster Discrete Convex Function Minimization with Predictions: The M-Convex Case"
_NeurIPS.cc/2023/Conference — NeurIPS 2023 poster_

### Official Review · Reviewer_xsf6 · 2023-07-04

**Soundness:** 2 fair
**Presentation:** 3 good
**Contribution:** 2 fair
**Rating:** 5
**Confidence:** 1

**Summary:**

In the vein of the seminal work by Sakaue and Oki for L-convex functions, this paper proposes a new method to accelerate M-convex function minimization using past predictions, a technique known as warm-start.

The contributions are as follows:
1) Present a framework to accelerate M-convex function minimization with past predictions
2) Applying this framework to the Laminar, Box and Nested classes of problems and improving time bounds using warm start for each of these problems

**Strengths:**

The results in the experiments section seems promising. Although I'm not an expert in discrete optimization, the problems tackled seemed of great importance.

**Weaknesses:**

1) This paper is very theoretically dense and tries to tackle a lot of problems at once, the contributions are not clear. I think this paper would gain at focusing solely on the Laminar problem and possibly present the extension of your work in appendix. The demonstration would be clearer, and your paper more straight to the point.

2) Although rigorously written, it feels like this paper is a re-adaptation of the work from Sakaue and Oki to the class of M-convex functions and is thus not particularly original.

3) The results section is rather light compared to the extent of the theoretical claims on the Laminar, Box and Nested subproblems. I'd have preferred for each of the 3 classes of problems you mention, a first experiment on synthetic data with comparison between warm start and no warm-start. Then, a second experiment on real-world data to prove the superiority of your method.

**Questions:**

- In figure 3, the y-axis label is set to "Iterations". What is the meaning of these iterations? Shouldn't this be the suboptimality gap?

**Limitations:**

The limitations are addressed.
No negative societal impact.

---

> ### Author Rebuttal · Authors · 2023-08-04
>
> We appreciate the reviewer's valuable comments. We are glad that the reviewer found our experimental results promising. Below, we respond to each comment.
>
> > **Weakness 1.** This paper is very theoretically dense and tries to tackle a lot of problems at once, the contributions are not clear. I think this paper would gain at focusing solely on the Laminar problem and possibly present the extension of your work in appendix. The demonstration would be clearer, and your paper more straight to the point.
>
> Firstly, we would like to clarify that our contribution is two-fold.
>
> 1. We present a general warm-start-with-prediction framework for M-convex minimization (as described in Theorem 3.1), and
> 2. based on Theorem 3.1, we obtain specific time complexity bounds shown in Table 1 for M-convex-minimization subclasses: Laminar, Nested, and Box.
>
> It should be noted that all our results can be interpreted in a unified manner as warm-starts with predictions for M-convex minimization. To demonstrate the versatility of this unified perspective, we have described how we can deal with those problems.
>
> As the reviewer noted, the result for Laminar is technically the most significant (with the result for Nested following as a special case of Laminar). However, as outlined in Section 1.1, our result for Box is also crucial as it is the first demonstration in the literature [11, 7, 39, 35, 10] that the theoretical worst-case lower bound [19] can be surpassed using predictions.
>
> We would appreciate it if the reviewer could understand the unified perspective and the significance of each result.
>
> > **Weakness 2.** Although rigorously written, it feels like this paper is a re-adaptation of the work from Sakaue and Oki to the class of M-convex functions and is thus not particularly original.
>
> Our work is not a re-adaptation of (Sakaue and Oki, 2022). Although L-convex minimization, addressed by Sakaue and Oki (2022), and M-convex minimization are nominally similar, they represent completely different classes of problems (L-convex encompasses bipartite matching, min-cost flow, etc., while M-convex includes Laminar, Nested, Box, etc.), hence named and studied separately in discrete convex analysis [31]. Consequently, our algorithms and theoretical analyses differ significantly from (Sakaue and Oki, 2022). Please refer to our [global response](https://openreview.net/forum?id=xtQ9IGRzIW&noteId=3iQVwbCgqD) for further details.
>
> > **Weakness 3.** The results section is rather light compared to the extent of the theoretical claims on the Laminar, Box and Nested subproblems. I'd have preferred for each of the 3 classes of problems you mention, a first experiment on synthetic data with comparison between warm start and no warm-start. Then, a second experiment on real-world data to prove the superiority of your method.
>
> We appreciate your suggestion. We have conducted additional experiments on realistic settings provided in [47]. Please refer to the [global response](https://openreview.net/forum?id=xtQ9IGRzIW&noteId=3iQVwbCgqD) for details. As with the experimental results in Section 5, under realistic settings (referred to as "CRASH" and "FUEL") with moderate noises, our method, Learn, outperformed the continuous-relaxation-based method, Relax. We hope the additional experimental results effectively address the reviewer's concern about the shortage of experiments.
>
> To clarify, as explained in lines 222--224, the Nested and Box problems are special cases of the Laminar problem. Thus, our experimental results for the Laminar problem could imply similar results for Nested and Box. (Indeed, the additional experimental results with Nested instances were similar to the original ones with Laminar instances.)
>
> > **Question.** In figure 3, the y-axis label is set to "Iterations". What is the meaning of these iterations? Shouldn't this be the suboptimality gap?
>
> By "Iterations," we refer to the number of iterations of the greedy algorithm (Algorithm 1) taken to solve each $t$th instance, as noted in the figure caption. Thus, the experimental result should read as follows: as the x-axis increases (indicating an increase in the number of observed past instances), the prediction tends to become more accurate (as implied in Proposition 3.2), reducing the number of iterations of the greedy algorithm required to solve a new instance (i.e., the y-axis). This experimental result validates our $\\|x^* - \hat x\\|_1$-dependent time complexity bounds, i.e., the time complexity decreases as we learn a prediction $\hat x$ from more past instances in order to approximate an optimal solution $x^*$ more accurately in expectation.

---

> > ### Comment · Reviewer_xsf6 · 2023-08-13
> >
> > I would like to thank the authors for their comprehensive answer that addressed the points made in my initial review. Although I'm not fully convinced by the novelty of this approach, I raised my evaluation from 3 to 5.

---

> > > ### Author Response · Authors · 2023-08-14
> > > **Gratitude for Your Feedback and Re-evaluation**
> > >
> > > We sincerely appreciate your feedback and the effort you devoted to re-evaluating our work. We are genuinely grateful for the improved score.
> > >
> > > We acknowledge that the novelty of our work, especially when juxtaposed with (Sakaue and Oki, 2022), might appear subtle and perhaps challenging to grasp. That being said, our algorithms and analyses indeed differ from the existing ones. They are pivotal in obtaining our time complexity bounds, which serve as significant demonstrations of the power of predictions for improving algorithms' performance beyond the worst-case limitations.
> > >
> > > We once again deeply appreciate your dedicated engagement with our work. Our gratitude cannot be overstated. Should there be any lingering questions or concerns, please do not hesitate to share them with us.

---

### Official Review · Reviewer_SFP3 · 2023-07-05

**Soundness:** 4 excellent
**Presentation:** 3 good
**Contribution:** 2 fair
**Rating:** 6
**Confidence:** 3

**Summary:**

This paper applies the learning-augmented algorithms framework to a class of discrete optimization problems called M-convex. A function defined on an integer grid is M-convex if for every x, y it holds that f(x)+f(y) >= f(x-ei+ej)+f(y+ei-ej) for some base vectors ei, ej. The paper complements the line of research on improving running time of static algorithms using predictions, started by Dinitz et al. (NeurIPS’21), and is a natural followup to the Sakaue&Oki’s paper (NeurIPS’22) on another class, called L-convex.

The authors propose a general framework for solving M-convex optimization problems given a predicted optimal solution that is supposed to be close to the true one. Then, they give algorithms for specific subclasses of these problems, called Laminar, Nested, and Box (after the form of constraints allowed in each class). The running times they obtain are O(n*eta) for Laminar and Nested and O(n+log(n)*eta) for Box, where eta = ||x_opt - x_pred||_1 is the L1-error of the prediction.

**Strengths:**

There are nice algorithms in Sections 4.1 and 4.2 (for rounding to the nearest feasible solution and for finding the steepest descent in the laminar case).

With (very) accurate predictions, the algorithm for the special case of box constraints can go (a tiny bit) below a known lower bound for classic (prediction-less) algorithms, and it seems to be the first such example in the literature about learning-augmented static algorithms.

The paper is written with care and not hard to follow.

**Weaknesses:**

The algorithm/framework for general M-convex optimization is technically trivial. The algorithms for the laminar subclass are much more interesting, but I’m not convinced that this subclass is interesting enough on its own for the results to have substantial implications.

Interesting technical ideas seem very much tailored to the specific setup, so it might be difficult for this paper to have a larger impact or inspire further research.

The experiments are very simplistic – on a toy problem with synthetic data and predictions. (Still it was somewhat surprising to learn that, for a concentrated enough distribution of instances, an integral solution learned from previous instances can be a better initialization than an optimal solution to the fractional relaxation.) If I consider this paper to be a pure theory paper (which for now I do), then such experiments are not an issue, but in that case the theoretical contributions of the paper might not be strong enough to pass the ICML bar.

**Questions:**

Could you find some real-world applications of laminar convex optimization that have existing benchmarks/datasets, and run experiments on them? (Even with some synthetic method for generating predictions.)

Statement in lines 59–60 seems not accurate. For instance, the negative-weight SSSP algorithm of [7] runs in O(m) time for very good predictions, beating the best known worst-case bound by a couple of log factors.

I do not agree with the statement in line 390 that you can beat best known algorithms “(...) given accurate predictions, which we can provably learn from past data.” The learnability proofs tell you that you can efficiently generate predictions as good as possible in a given context, but not that there exist predictions with small enough error. (This is a common limitation of PAC-learning framework, so I’m not complaining about your result, just about the way in which you describe it.)

Minor comments:

Line 26: At first it was totally unclear to me what “conjugate” was supposed to mean in this context.

Line 28: “Specifically, we focus on an important subclass (...)” – this sentence is misleading as it suggests that you only focus on the subclass, which is not the case, because you address the general problem as well.

Line 30: “(...) is widely studied (...)” – could you please provide some references?

Line 42: Please mention that the greedy algorithm is given later as Algorithm 1.

Line 101: “closet” -> “closest”

Line 223: For readability, I’d consider using a colon instead of \mid because \mid blends with the set cardinality symbol.

**Limitations:**

Treating it as a theory paper, it does not have any important limitations (apart from the possibly limited impact itself). The title might be slightly misleading, since the most interesting results are only for a relatively narrow special case (laminar) and not the general case mentioned in the title (L-convex).

---

> ### Author Rebuttal · Authors · 2023-08-04
>
> We appreciate the reviewer's detailed and constructive comments. We are pleased that the reviewer recognizes the technical strengths of the algorithms in Sections 4.1 and 4.2 and the significance of our result for Box, a first demonstration of the potential to surpass the lower-bound result using predictions. Below are our responses to the review comments.
>
> ### On weaknesses
> > I'm not convinced that this subclass is interesting enough on its own for the results to have substantial implications.
>
> Although the subclasses (the Laminar, Nested, and Box problems) might not be as widely recognized in the machine learning (ML) community as the ones addressed in the literature (Dinitz et al. [11]; Chen et al. [7]; Davis et al. [10]; Polak and Zub [35]), those problems are extensively studied in operations research (OR) [18, 19, 20, 22, 30, 41, 42, 43, 46, 47] for their importance in the industrial domain. We believe our work can deliver a substantial impact by catalyzing cross-disciplinary future research between ML and OR.
>
> > Interesting technical ideas seem very much tailored to the specific setup, so it might be difficult for this paper to have a larger impact or inspire further research.
>
> Although the individual algorithms in Section 4 may appear specific, they stem from our general framework M-convex minimization in Section 3. This general framework tells us that efficient $\ell_1$-projection and computation of a steepest descent direction of the form $-e_i + e_j$ are crucial for using predictions to accelerate algorithms for the broad class of M-convex minimization. This general implication could benefit future research by providing a clear direction for enhancing M-convex minimization algorithms with predictions.
>
> ### On questions
> > Could you find some real-world applications of laminar convex optimization that have existing benchmarks/datasets, and run experiments on them? (Even with some synthetic method for generating predictions.)
>
> In response to the question, we conducted additional experiments using Nested instances from [47]. Please refer to our [global response](https://openreview.net/forum?id=xtQ9IGRzIW&noteId=3iQVwbCgqD) for details. (Unfortunately, we could not find datasets for Laminar instances due to the scarcity of public datasets in this domain.) Under realistic settings (referred to as "CRASH" and "FUEL") with moderate noises, our method, Learn, outperformed the continuous-relaxation-based method, Relax, as with the results in Section 5. We believe these additional experiments strengthen the practical side of our work and hope the reviewer will appreciate them.
>
> > Statement in lines 59–60 seems not accurate. For instance, the negative-weight SSSP algorithm of [7] runs in O(m) time for very good predictions, beating the best known worst-case bound by a couple of log factors.
>
> We thank the reviewer for pointing this out. We will revise lines 59--60 to clarify that Chen et al. [7] 's SSSP algorithm with predictions can surpass the best worst-case bound.
>
> We would like to make two remarks to ensure that all reviewers have a correct understanding. (i) We are still the first to demonstrate the potential to surpass the *lower-bound* result [19] using predictions. (ii) Given accurate predictions, our algorithm for Laminar can achieve an $\mathrm{O}(n)$-factor improvement over the best-known algorithm [18, 34], in contrast to Chen et al. [7] 's logarithmic improvement upon the best algorithm.
>
> > I do not agree with the statement in line 390 that you can beat best known algorithms "(...) given accurate predictions, which we can provably learn from past data." The learnability proofs tell you that you can efficiently generate predictions as good as possible in a given context, but not that there exist predictions with small enough error. (This is a common limitation of PAC-learning framework, so I'm not complaining about your result, just about the way in which you describe it.)
>
> We appreciate this comment. As the reviewer mentioned, there is an inevitable limitation of agnostic PAC learning, i.e., even optimal predictions may not achieve small errors depending on the situation. We will clearly state this general limitation of the agnostic PAC learning framework in our revision.
>
> We also appreciate the minor comments and will revise our title to better reflect our primary findings, as suggested.

---

> > ### Comment · Reviewer_SFP3 · 2023-08-11
> >
> > Thanks for the response. Given the above explanations about importance of M-convex optimization in OR literature (and the comments of reviewer KCcb, who seem to be more knowledgeable in the area), as well as the additional experiments, I increase my score to 6.

---

> > > ### Author Response · Authors · 2023-08-14
> > > **Gratitude for Your Feedback and Re-evaluation**
> > >
> > > We deeply appreciate your meticulous attention to our paper, both in your initial detailed feedback and your subsequent review of our rebuttal. We are genuinely grateful for the revised score and heartened to know that the importance of our work in the OR literature has been recognized. We truly value your contributions to the review process. Should there be any lingering questions or concerns, we warmly invite you to share them with us.

---

### Official Review · Reviewer_KCcb · 2023-07-08

**Soundness:** 4 excellent
**Presentation:** 4 excellent
**Contribution:** 3 good
**Rating:** 7
**Confidence:** 4

**Summary:**

The goal of the work is to minimize M-convex functions with prediction. The work mainly focuses on a subclass of M-convex functions that use Laminar, Nested or Box constraints.

**Strengths:**

Minimizing M-convex functions is an important class of discrete optimization problems that has wide variety of applications. Using algorithms with prediction is an important area of research that can typically solve many problems in discrete convex analysis. Especially due to wide variety of application.

Soundness. The work is technically sound that uses relevant algorithms produced in literature to solve a very important problem.

Presentation. The paper is very well written that flows flawlessly with very minor concerns.

Contribution. The contribution is first of its kind to the class of problems that it is applied it. However, it must be noted that most of the major algorithms used in the paper are proposed in literature. The main contribution is modification of these classical algorithms to solve the bigger problem of minimizing M-convex functions with predictions.

**Weaknesses:**

I dont find many weaknesses in this work.

**Questions:**

I have two major concerns with the work

1. In section 4.2, it is clear that finding the node $j$ for a fixed node $i$ can be done in $O(n)$ time. However, I think it is not  $O(n)$ as the Dynamic programming algorithm iterates over $\mathcal{C}(Y)$ and this is not a constant which depends on the type of laminar function. On the other hand, such a tree is much simpler for a nested case or box case.

2. For the sake of clarity, I request the authors to also explain the learning predictions part even though it is the same as used in other seminal works at least in the appendix. The paper is not self contained otherwise.

**Limitations:**

This work would have been more complete if more efficient algorithms could be proposed for submodular constraints. This is the only reason for me to not give strong accept. Although laminar, nested and box are interesting, I believe, they have limited application scope.

---

> ### Author Rebuttal · Authors · 2023-08-08
>
> We appreciate the reviewer's careful reading and constructive feedback. We are pleased that the reviewer understood our main contribution and appreciated the importance of our results. Below are our responses to the comments.
>
> > **Question 1.** In section 4.2, it is clear that finding the node for a fixed node can be done in $O(n)$ time. However, I think it is not $O(n)$ as the Dynamic programming algorithm iterates over $\mathcal{C}(Y)$ and this is not a constant which depends on the type of laminar function. On the other hand, such a tree is much simpler for a nested case or box case.
>
> We assume, for simplicity, that the tree $T_\mathcal{F}$ is binary, i.e., $|\mathcal{C}(Y)| \le 2$, as stated in lines 231--233. Thus, the dynamic programming algorithm terminates in $\mathrm{O}(n)$ time. It should be noted that we can always preprocess $T_\mathcal{F}$ to make it binary by adding at most $n$ dummy nodes without worsening the asymptotic time complexity. This treatment is indeed only for simplifying the analysis. Even if $T_\mathcal{F}= (\mathcal{V}, E)$ is non-binary, the sum of $|\mathcal{C}(Y)|$ over all nodes $Y \in \mathcal{V}$ is $n - 1$ since every non-root node has exactly one parent. Thus, the total time complexity of the dynamic programming algorithm is $\mathrm{O}(\sum_{Y \in \mathcal{V}} |\mathcal{C}(Y)|) = \mathrm{O}(n)$ anyway.
>
> > **Question 2.** For the sake of clarity, I request the authors to also explain the learning predictions part even though it is the same as used in other seminal works at least in the appendix. The paper is not self contained otherwise.
>
> We appreciate the reviewer's helpful suggestion. We agree with the need to elaborate further on the learning of predictions for completeness, although the procedure is the same as (Khodak et al., 2022). We will include a more detailed explanation in our revision.
>
> > **Limitation.** This work would have been more complete if more efficient algorithms could be proposed for submodular constraints. This is the only reason for me to not give strong accept. Although laminar, nested and box are interesting, I believe, they have limited application scope.
>
> We appreciate this comment. Certainly, developing more efficient algorithms for general submodular constraints is of great significance; this is an avenue for future work. Below is our perspective on this challenge. In the general M-convex case, we obtain an initial feasible solution by projecting a prediction onto the base polyhedron of a submodular function. This process contains the membership testing for the base polyhedron as a special case, which is considered to be as hard as submodular minimization due to the equivalence of minimization and separation. Therefore,  designing a projection algorithm that is much faster than submodular minimization (or $\mathrm{O}(\textsf{SFM})$-time) for general M-convex minimization would be challenging.
>
> On the other hand, our $\mathrm{O}(n\textsf{SFM})$-time projection described in Section 3.1 and Appendix A.1 is helpful in that it ensures the existence of a polynomial-time projection algorithm for general M-convex minimization. That is, when we consider applying our warm-start-with-prediction framework to new M-convex minimization subclasses, we know from our general $\mathrm{O}(n\textsf{SFM})$-time bound that the projection can be done in polynomial time. This information effectively focuses our attention on designing more efficient projection algorithms using problem-specific structures.

---

> > ### Comment · Reviewer_KCcb · 2023-08-17
> >
> > I thank the authors for their detailed feedback. I have gone through the other reviews too and the effort the authors put to answer all the questions is commendable. I intend to stick to my rating that I provided earlier.

---

> > > ### Author Response · Authors · 2023-08-18
> > > **Gratitude for Your Feedback**
> > >
> > > We sincerely appreciate the reviewer's thoughtful feedback and careful examination of other reviews. Your constructive engagement in the review process has truly encouraged us. We are delighted with your agreement towards acceptance.

---

### Official Review · Reviewer_NNSF · 2023-07-24

**Soundness:** 3 good
**Presentation:** 3 good
**Contribution:** 2 fair
**Rating:** 5
**Confidence:** 3

**Summary:**

Extending the warm-starting techniques in L-convex function minimization by Sakaue and Oki (2022), the authors study the problem of acclerating M-convex function minimization with predictions. The idea is to start from a (possibly infeasible) predicted solution, project the rounded solution to the feasible region and then apply the standard M-convex function minimization greedy algorithm. In particular, the authors show that when applied to Laminar convex minimization, a special case of M-convex function minimization, their framework can achieve better time complexity than current worst-case time complexity provided that the prediction is accurate enough. Experiments on the staff-assignment problem confirm that the proposed framework can help reduce the number of iterations of the greedy algorithm.

**Strengths:**

The paper studies an interesting problem, and is written well in terms of structure, clarity in explanation and technical presentation. Theoretical results are justified by empirical experiments.

**Weaknesses:**

1. The framework proposed in this paper for M-convex optimization is somewhat similar to the one in Sakaue and Oki (2022) for L-convex optimization. So novelty in the general framework is limited.
2. The authors provide comparision between their results and worst-case time complexity for laminar convex minimization. However, it seems like their complexity outperforms the worst-case only under the assumption of highly accurate predictions (small enough $\ell_1$ norm prediction error), which does not necessarily hold in practice.

**Questions:**

The staff assignment problem does not seem to be a very realistic test problem to me (with 12800 staff members and 128 tasks). Are there more standard test problems? How does the proposed framework perform on them?

**Limitations:**

The limitations have been addressed.

---

> ### Author Rebuttal · Authors · 2023-08-05
>
> We are grateful to the reviewer for providing valuable comments. Below we respond to each comment.
>
> > **Weaknesses 1.** The framework proposed in this paper for M-convex optimization is somewhat similar to the one in Sakaue and Oki (2022) for L-convex optimization. So novelty in the general framework is limited.
>
> While our framework for M-convex minimization, at a high level, resembles that for L-convex minimization in (Sakaue and Oki, 2022), these two classes are entirely different, as detailed in our [global response](https://openreview.net/forum?id=xtQ9IGRzIW&noteId=3iQVwbCgqD). In particular, the definitions of convexity and corresponding steepest descent directions completely differ between L- and M-convex cases. Thus, new efficient methods for projection and computation of steepest directions are required, as discussed in Section 4. We hope the reviewer understands that our paper has sufficient novelty, despite the apparent similarity of the general frameworks.
>
> > **Weaknesses 2.** The authors provide comparision between their results and worst-case time complexity for laminar convex minimization. However, it seems like their complexity outperforms the worst-case only under the assumption of highly accurate predictions (small enough $\ell_1$ norm prediction error), which does not necessarily hold in practice.
>
> Assuming accurate predictions in order to surpass the worst-case limitations is standard in the literature [11, 7, 39, 35, 10]; similar ideas are also common in the *beyond-the-worst-case* paradigm [38]. Researchers in these fields have theoretical interests in doing better than the worst-case limitations on algorithm performance by leveraging past data. Therefore, even if accurate predictions are demanded, our theoretical time complexity bounds, which potentially surpass the best worst-case results [14, 18, 19, 34, 46] and a lower-bound result [19], are of great significance. It should also be noted that our $\mathrm{O}(n\\| x^* - \hat x\\|_1)$-time bound for the Laminar problem outperforms the best known bound of $\mathrm{O}(n^2\log n \log (mR/n))$ [18, 34] if $\\| x^* - \hat x\\|_1 = \mathrm{O}(n)$, i.e., every element is allowed to have a constant error on average. Surpassing the best worst-case results with such a mild assumption on prediction accuracy is rare in this context.
>
> We also remark that if accurate predictions do not exist, surpassing the worst-case limitations is inevitably difficult. That is, if past instances exhibit no tendency, we cannot utilize them to improve algorithm performance. Considering this, as in Proposition 3.2, we aim to learn predictions that perform best on the underlying distribution of instances, hoping that such best predictions incur small errors in practice. This idea is customary in general *agnostic PAC learning*, as also mentioned by Reviewer [SFP3](https://openreview.net/forum?id=xtQ9IGRzIW&noteId=ci542rY7dc). We will clearly state this point in our revision.
>
> > **Question.** The staff assignment problem does not seem to be a very realistic test problem to me (with 12800 staff members and 128 tasks). Are there more standard test problems? How does the proposed framework perform on them?
>
> In response to the reviewer's question, we conducted additional experiments using Nested instances from [47]. Please refer to our [global response](https://openreview.net/forum?id=xtQ9IGRzIW&noteId=3iQVwbCgqD) for details. As with the results in Section 5, under realistic settings (referred to as "CRASH" and "FUEL"), our method, Learn, outperformed the continuous-relaxation-based method, Relax, in scenarios with moderate noises. We believe that the additional experiments adequately validate the performance of our method in practice and hope the reviewer will appreciate them.
>
> > **Limitation.** The authors admit that solving real-world instances requires tailored methods for learning predictions. Therefore, it is not clear when one can benefit from the framework the authors propose.
>
> We would like to clarify that tailored learning methods are not always necessary. Predictions learned via standard online subgradient descent, as in our experiments, are helpful enough to reduce the computation cost of solving real-world instances, which the additional experiments also validate. We intended to express that such tailored learning methods could further enhance performance. We apologize for the misleading description.

---

> > ### Comment · Reviewer_NNSF · 2023-08-10
> >
> > I appreciate your response and the new sets of experiments. They certainly contribute to a better understanding of the issues I had raised.

---

> > > ### Author Response · Authors · 2023-08-11
> > > **Gratitude for Your Feedback**
> > >
> > > We sincerely appreciate your swift feedback and the effort dedicated to the review process. It's a privilege to interact with a responsive and considerate reviewer like you. We're pleased to know that our response and additional experiments have addressed your concerns.
> > >
> > > In accordance with the reviewer guidelines, which mention, "If your evaluation of the paper has changed, please revise your review and explain the change," we kindly request that you consider updating your review if our response has affected your evaluation. Should you have any further questions or concerns, please do not hesitate to share them with us.

---

### Official Review · Reviewer_pZm4 · 2023-07-25

**Soundness:** 3 good
**Presentation:** 3 good
**Contribution:** 2 fair
**Rating:** 7
**Confidence:** 4

**Summary:**

This paper studies some classes of $M$-convex minimization problems, to which the recent framework of "warm-starts with predictions" is applied. The paper provides provable time complexity bounds on the standard greedy algorithm for $M$-convex minimization where the bounds are dependent on the $\ell_1$-distance between the optimal solution and the predicted initial solution. The theoretical performance guarantees are promising and improve upon the existing methods that are not using predictions. At the same time, however, one would argue that the technical contributions of this paper are limited in that the results rely on and are deduced by applications of the existing results in the literature. Furthermore, one would be interested to see the numerical impact of the framework on more concrete problem settings, e.g., portfolio management and resource allocation, which were not tested in this paper.

**Strengths:**

* The time complexity bounds provided in this paper are the first results that analyze the performance of the framework of warm-starts with predictions applied to $M$-convex function minimization.

* The time complexity bounds improve upon the existing bounds for some classes of $M$-convex minimization problems when we may obtain an accurate prediction where the accuracy is measured by the $\ell_1$-distance to the (unique) optimal solution.

**Weaknesses:**

* ​The technical contributions of this paper are limited. The greedy algorithm and main results of this paper are built upon and follow from [(Shioura (2022), Corollary 4.2)](https://pubsonline.informs.org/doi/abs/10.1287/moor.2021.1180) which gives an upper bound on the number of required iterations for the greedy algorithm in terms of the proximity term. The rest of the results are basically about bounding $T_{\text{init}}$, the time required to convert a prediction to a feasible solution, and $T_{\text{loc}}$, the time bound for computing a locally steepest direction. Even these results follow from standard techniques in the literature.

* This paper lacks computational demonstration. $M$-convex minimization has applications in resource allocation, equilibrium analysis, and portfolio management, but none of these problems were tested. In particular, one would be interested in how well the framework of this paper performs for the operations management models studied in [(Chen and Li, (2021))](https://pubsonline.informs.org/doi/abs/10.1287/opre.2020.2070). Furthermore, the numerical results reported in Section 6 do not consider the methods against which the theoretical complexity bounds are compared.

**Questions:**

* When testing the performance of this paper's framework against the methods without predictions, would it be fair to consider the time to compute a good prediction? Proposition 3.2 provides the regret bound by the standard online learning method, but what would be the required number of iterations to deduce the desired proximity bound on $\|x^*-\hat x\|_1$? Can you compare the total time required by the framework of this paper, adding up the time for learning a good prediction and the time to solve the problem with the warm start, against the methods without predictions?

* You mention that the required framework for $M^\natural$-convex minimization is similar to that for $M$-convex minimization. Can you provide more details?

**Limitations:**

The authors have adequately addressed the limitations.

---

> ### Author Rebuttal · Authors · 2023-08-08
>
> We appreciate the reviewer's insightful comments. We are delighted that the reviewer has found our improvements using predictions upon existing methods promising. We respond to each comment below.
>
> ### On weaknesses
> > ​The technical contributions of this paper are limited. The greedy algorithm and main results of this paper are built upon and follow from (Shioura (2022), Corollary 4.2) which gives an upper bound on the number of required iterations for the greedy algorithm in terms of the proximity term. The rest of the results are basically about bounding $T_{\text{init}}$, the time required to convert a prediction to a feasible solution, and $T_{\text{loc}}$, the time bound for computing a locally steepest direction. Even these results follow from standard techniques in the literature.
>
> While deriving the general bound (Theorem 3.1) from (Shioura (2022), Corollary 4.2) is not difficult, our techniques used in Section 4 for bounding $T_{\text{init}}$ and $T_{\text{loc}}$ are not straightforward, as detailed in [global response](https://openreview.net/forum?id=xtQ9IGRzIW&noteId=3iQVwbCgqD). Specifically, we have modified the $\mathrm{O}(n\log^2 n)$-time convolution (Teng and Luo, 1996) to achieve $T_{\text{init}} = \mathrm{O}(n)$-time projection, as in Section 4.1. Furthermore, we have improved the direction-finding method of (Moriguchi et al. 2011) to achieve $T_{\text{loc}} = \mathrm{O}(n)$-time direction finding (where the direct use of the original one results in $T_{\text{loc}} = \mathrm{O}(n^2)$, as discussed in lines 311--313). Moreover, our polynomial-time projection and direction-finding methods for the general $\text{M}$-convex case require non-trivial techniques on the base polyhedron of submodular functions, as discussed in Section 3.1 and Appendix A.1.
>
> > This paper lacks computational demonstration.
> $\text{M}$-convex minimization has applications in resource allocation, equilibrium analysis, and portfolio management, but none of these problems were tested. In particular, one would be interested in how well the framework of this paper performs for the operations management models studied in (Chen and Li, (2021)). Furthermore, the numerical results reported in Section 6 do not consider the methods against which the theoretical complexity bounds are compared.
>
> We conducted additional experiments using Nested instances from [47], which involve two real-world settings, project crashing (CRASH) and ship speed optimization (FUEL), and one synthetic setting (F). Please refer to our [global response](https://openreview.net/forum?id=xtQ9IGRzIW&noteId=3iQVwbCgqD) for details. Under the CRASH and FUEL settings with moderate noises, our method, Learn, outperformed the continuous-relaxation-based method, Relax, as with the results in Section 5. We believe these additional experimental results strengthen the practical side of our work and hope the reviewer will appreciate them.
>
> Unfortunately, we could not find public datasets for operation management models in (Chen and Li, 2021). Also, we employed the continuous-relaxation method (Relax) rather than the theoretically fast methods for the convenience of fair comparisons based on the number of iterations of the greedy algorithm, as detailed in [global response](https://openreview.net/forum?id=xtQ9IGRzIW&noteId=3iQVwbCgqD).
>
> ### On questions
> > When testing the performance of this paper's framework against the methods without predictions, would it be fair to consider the time to compute a good prediction? Proposition 3.2 provides the regret bound by the standard online learning method, but what would be the required number of iterations to deduce the desired proximity bound on? Can you compare the total time required by the framework of this paper, adding up the time for learning a good prediction and the time to solve the problem with the warm start, against the methods without predictions?
>
> In the literature [11, 7, 39, 35, 10], it is customary and regarded as fair to consider only the time to solve new instances when comparing against the methods without predictions; i.e., we assume predictions are given. This is because data of past instances is already provided, and learning a prediction $\hat x$ from it is usually allowed to take much longer than the time for solving an upcoming instance. That is, only $T_\text{init} + T_\text{loc}\\| x^* - \hat x\\|_1$ matters when we focus on solving upcoming instances quickly, and we learn $\hat x$ beforehand to make $\\| x^* - \hat x\\|_1$ small.
>
> Furthermore, in our case, predictions are learned via online subgradient descent whose *single* iteration constitutes the time for learning a prediction at each $t$ (see lines 369--371). This increase in the total computation time is negligible compared to the hundreds of iterations of the greedy algorithm (Algorithm 1) on each $t$th instance.
>
> > You mention that the required framework for $\text{M}$-convex minimization is similar to that for $\text{M}^\natural$-convex minimization. Can you provide more details?
>
> There is a one-to-one correspondence between $\text{M}^\natural$-convex functions of $n$ variables and $\text{M}$-convex functions of $n+1$ variables, as described in Section 6.1 in [31]. Indeed, for an $\text{M}^\natural$-convex function $f: \mathbb{Z}^N \to \mathbb{R} \cup \{+\infty\}$, we can construct an $\text{M}$-convex function $\tilde{f}: \mathbb{Z} \times \mathbb{Z}^N \to \mathbb{R} \cup \{+\infty\}$ by setting $\tilde{f}(x_0, x) = f(x)$ if $x_0 = -x(N)$ and $\tilde{f}(x_0, x) = +\infty$ otherwise for $x_0 \in \mathbb{Z}$ and $x \in \mathbb{Z}^N$. Clearly, $x \in \mathbb{Z}^N$ minimizes $f$ if and only if $(-x(N), x)$ minimizes $\tilde{f}$.
> Thus, we can apply our warm-start framework to $\text{M}$-convex $\tilde{f}$ to solve the original $\text{M}^\natural$-convex minimization of $f$. Alternatively, we can slightly modify our framework to deal with $\text{M}^\natural$-convex minimization directly without using $\tilde{f}$.

---

> > ### Comment · Area_Chair_ieDo · 2023-08-21
> >
> > As the reviewer did not yet respond, to the author response, I will provide additional information and feedback.
> >
> > Regarding the technical novelty. I believe the author response to be satisfactory. The response regarding establishing the linear worst-case time complexity dispels the reviewer's comment.
> >
> > Regarding additional experiments, I remain neutral. I believe the NESTED case adds additional results strengthening the claims, but I do agree with the reviewer's comments about wider study of application. However, as these experiments are comparable to previous work I am familiar in this area, I do not believe this to be an overly negative factor for the paper.
> >
> > I believe the response to the questions is also helpful.
> >
> >
> > Thank you

---

> > > ### Author Response · Authors · 2023-08-21
> > > **Gratitude to the Area Chair for Valuable Feedback**
> > >
> > > We sincerely thank the Area Chair for their insightful feedback on our paper. It's reassuring to know that the Area Chair has understood our response addressing the reviewer's comment on the technical novelty. Regarding the experiments, we understand that even with the additions, there remains room for improvement. Still, we believe that they effectively complement the practical side of our theoretical findings. Once again, our genuine appreciation goes out to the Area Chair's unwavering dedication to the review process and valuable feedback on our work.

---

> > ### Comment · Reviewer_pZm4 · 2023-08-21
> >
> > Thank you so much for the detailed comments on my earlier review report and apologies for my late response. I agree with the area chair's judgment that the paper has more technical significance than my earlier evaluation. I also appreciate the additional numerical experiments conducted for a short period of time. That said, I raised my score from 6 to 7.

---

> > > ### Author Response · Authors · 2023-08-21
> > > **Gratitude for Your Feedback and Re-evaluation**
> > >
> > > We deeply appreciate the reviewer's valuable feedback. We fully understand that the discussion period is often extremely busy. We are truly grateful that despite this, you took the time to review our response and additional experiments, provide feedback, and revise the score. Thank you very much.

---

### Official Review · Reviewer_xgTQ · 2023-07-26

**Soundness:** 2 fair
**Presentation:** 2 fair
**Contribution:** 2 fair
**Rating:** 5
**Confidence:** 1

**Summary:**

The paper discusses the growing interest in accelerating optimization algorithms using machine-learned predictions. It highlights the work of Sakaue and Oki, who introduced a general framework for employing predictions to warm-start the L-convex function minimization method, demonstrating its effectiveness for various discrete optimization problems. Building on this, the paper presents a new framework that leverages predictions to accelerate M-convex function minimization with improved time complexity bounds, thereby extending the applicability of predictive techniques to a wider range of discrete optimization algorithms.

**Strengths:**

This paper is well-written, presenting intuitive results. It exhibits technical solidity, providing clear explanations of how the proposed algorithm enhances the existing worst-case time complexity.

**Weaknesses:**

1. The main theoretical results appear to be straightforward. The computational time complexity relies on the distance between the initialization and the optimal solution, assuming a good initialization is provided. However, one limitation of this paper is the absence of an analysis regarding the methodology for obtaining such a good initialization.
2. The difficulty in deriving the corresponding theoretical results is not well elucidated in the paper. For instance, proving Theorem 3.1 hinges on Proposition 2.2, which is simply a direct application from existing works [44]. It would be more beneficial if the paper provided a clearer explanation of the technical challenges involved in obtaining the theoretical results.
3. This paper lacks numerical justifications for the proposed theoretical time complexity bound.


**Questions:**

Could the authors present concrete evidence illustrating how the theoretical bounds offer valuable insights capable of significantly accelerating the practical applications using the proposed algorithm?

**Limitations:**

Yes

---

> ### Author Rebuttal · Authors · 2023-08-05
>
> We thank the reviewer for providing valuable feedback. We present our response to each comment below.
>
> > **Weakness 1.** The main theoretical results appear to be straightforward. The computational time complexity relies on the distance between the initialization and the optimal solution, assuming a good initialization is provided. However, one limitation of this paper is the absence of an analysis regarding the methodology for obtaining such a good initialization.
>
> We would like to clarify that our contribution is not merely deriving the general distance-dependent bound (Theorem 3.1). More important results are specific time complexity bounds for the Laminar, Nested, and Box problems (Table 1), which are significant in the literature [11, 7, 39, 35, 10] as they demonstrate the possibility of surpassing the best worst-case bounds and even a lower bound [19] using predictions, as discussed in Section 1.1. To achieve those bounds, we have developed fast projection and direction-finding methods and combined them with Theorem 3.1, as discussed in Section 4. Please refer to our [global response](https://openreview.net/forum?id=xtQ9IGRzIW&noteId=3iQVwbCgqD) for further details of our technical novelty for developing those algorithms.
>
> We also point out that the procedure for obtaining a good initialization (or learning a prediction) is covered in lines 169--187. As we have discussed there, we can use the same online subgradient descent method as [23] for our purpose, which enjoys the regret and sample-complexity bounds as stated in Proposition 3.2.
>
> > **Weakness 2.** The difficulty in deriving the corresponding theoretical results is not well elucidated in the paper. For instance, proving Theorem 3.1 hinges on Proposition 2.2, which is simply a direct application from existing works [44]. It would be more beneficial if the paper provided a clearer explanation of the technical challenges involved in obtaining the theoretical results.
>
> We would like to re-emphasize that Theorem 3.1 is not our primary result. Our main results are the time complexity bounds for the Laminar, Nested, and Box problems, summarized in Table 1, and the core technical challenges reside in devising efficient algorithms for those problems, as described in Section 4.
>
> > **Weakness 3.** This paper lacks numerical justifications for the proposed theoretical time complexity bound.
>
> We conducted additional experiments using Nested instances from [47]. Please refer to our [global response](https://openreview.net/forum?id=xtQ9IGRzIW&noteId=3iQVwbCgqD) for details. Results from the additional experiments, along with the original ones in Section 5, indicate that by learning predictions $\hat x$ from more past instances to approximate optimal solutions $x^*$ better in expectation, we can reduce the number of iterations of the greedy algorithm, thus justifying our theoretical $\\|x^* - \hat x\\|_1$-dependent time complexity bounds.
>
> > **Question.** Could the authors present concrete evidence illustrating how the theoretical bounds offer valuable insights capable of significantly accelerating the practical applications using the proposed algorithm?
>
> The experiments in Section 5 and the additional ones in the [global response](https://openreview.net/forum?id=xtQ9IGRzIW&noteId=3iQVwbCgqD) offer concrete evidence that the computational cost decreases in practice as predictions become more accurate, aligning directly with our $\\|x^* - \hat x\\|_1$-dependent time complexity bounds. More precisely, the number of iterations of the greedy algorithm (y-axis) decreases as the number of past instances observed (x-axis) increases, where the increasing number of observed instances enables learning of predictions $\hat x$ closer to optimal solutions $x^*$ in expectation. In essence, our theoretical bounds offer a practical implication that learning better predictions on larger datasets accelerates the greedy algorithm, which the experiments validate empirically.
>
> As previously mentioned, Proposition 3.2 ensures that we can learn predictions $\hat x$ that are close to the best-performing prediction $\hat x^*$ given enough past instances. Specifically, we can use the online subgradient descent method to learn such predictions; please also refer to lines 369--372 for the learning procedure used in our experiments.

---

> > ### Comment · Reviewer_xgTQ · 2023-08-11
> > **Thanks for the clarification and additional experiments**
> >
> > I appreciate the authors' efforts in providing further clarification and conducting additional experiments. While it's possible that my own limited intuition and familiarity with the field could be contributing to this, I still find it challenging to grasp the technical novelty inherent in the theoretical results presented in this paper. Nevertheless, with the inclusion of supplementary experiments that validate the proposed theoretical findings, I have revised my initial score from 4 to 5.

---

> > > ### Author Response · Authors · 2023-08-14
> > > **Gratitude for Your Feedback and Re-evaluation**
> > >
> > > We sincerely appreciate your thoughtful reconsideration of our work, especially in light of the additional experiments we conducted. Your dedication to the review process is evident, and we are truly grateful for the improved score.
> > >
> > > We acknowledge that fully grasping the novelty of our research can pose challenges. For a complete understanding, one needs to delve into various existing techniques [30, 44, 45] and the recent study by Sakaue and Oki (2022). Nevertheless, we are confident in the significance of our results, especially in the context of "acceleration of algorithms with machine-learned predictions," established by the seminal work by Dinitz et al. [11]. Our experiments also corroborate the results.
> > >
> > > Once again, we deeply value your earnest engagement with our paper and your thoughtful response to our rebuttal. Should you have any further questions or concerns, please do not hesitate to share them with us.

---

### Author Rebuttal · Authors · 2023-08-04

# Global response on experiments and technical novelty
We sincerely thank all reviewers for providing valuable feedback. Given the mixed reviews, we deem it necessary to begin by addressing key comments. Below, we address comments on experiments and technical novelty.

## Experiments
First, we recap our experimental results. In Figure 3, the x- and y-axes indicate the numbers of past instances observed and of iterations taken by the greedy algorithm (Algorithm 1) to solve the new $t$th instance, respectively. Increasing x-axis values imply that predictions $\hat x$ better approximate optimal solutions $x^*$ in expectation (cf. Proposition 3.2). Our proposed method, denoted by "Learn," effectively reduces the number of iterations as the number of observed instances grows, supporting our $\\|x^* - \hat x\\|_1$-dependent time complexity bounds.

We also emphasize that the "Relax" benchmark method is a strong competitor, which follows the same continuous-relaxation idea as the fastest method for Laminar with quadratic objectives [30]. As Reviewer [SFP3](https://openreview.net/forum?id=xtQ9IGRzIW&noteId=ci542rY7dc) mentioned, it is noteworthy that our experiments have shown the potential to outperform Relax by using predictions. We could also implement and test other existing methods. However, while we are interested in how predictions affect the number of iterations of the greedy algorithm, other methods do not necessarily take the form of a similar iterative method, making a fair comparison difficult. Even if we were to compare based on running time, the results would be influenced by implementation and could not yield informative outcomes.

We conducted additional experiments to address the concern about the lack of experiments with standard datasets. While public datasets are scarce in this domain, we have found that Wu et al. [47] have made their code for the Nested problem available. As in [47, Section 6.3], there are three types of objective functions: "F," "CRASH," and "FUEL." F is a synthetic fourth-order polynomial, while CRASH and FUEL come from real-world project crashing and ship speed optimization, respectively. Following [47], we obtained Nested instances with those objectives and generated $T=100$ instances by perturbing parameters defining constraints and objectives with Gaussian noises multiplied by $\sigma = 0.1, 1.0, 10.0$, which controls the noise strength. As with Section 5, we measured the number of iterations of the greedy algorithm initialized by Cold (the cold-start baseline), Relax, and Learn over the $100$ instances.

The results are shown in the attached PDF file. For CRASH and FUEL with $\sigma = 0.1, 1.0$, Learn outperformed Relax, as with the results in Section 5. As for F, Relax performed extremely well and beat Learn, probably because the synthetic fourth-order objective is easy to handle for the continuous-relaxation method. Still, it is significant that Learn can surpass Relax for CRASH and FUEL, which come from real-world applications, under moderate noises. We believe these results effectively address reviewers' concerns about the performance of our method in practice.

## Technical novelty
Reviewers [NNSF](https://openreview.net/forum?id=xtQ9IGRzIW&noteId=Ae4ha8dz3k) and [xsf6](https://openreview.net/forum?id=xtQ9IGRzIW&noteId=CzMG1jdMzT) have expressed concerns about the perceived overlap with (Sakaue and Oki, 2022). Sakaue and Oki (2022) have studied warm-starts with predictions for L-convex minimization, and the high-level idea of using predictions for discrete convex minimization is similar to ours. However, M-convex minimization (which includes the Box, Nested, and Laminar problems addressed in our paper) is fundamentally different from L-convex minimization (which includes bipartite matching, min-cost flow, etc.). This is why they are named and studied separately in discrete convex analysis [31]. In particular, the definitions of feasible regions (or convex sets) and locally steepest directions entirely differ between M- and L-convex cases. Consequently, new methods for projection and finding directions for M-convex minimization are required, which we have developed for Laminar, Nested, and Box in Section 4.

Reviewers [xgTQ](https://openreview.net/forum?id=xtQ9IGRzIW&noteId=BpSA9YYtSV) and [pZm4](https://openreview.net/forum?id=xtQ9IGRzIW&noteId=2mlaXbCFul) have suggested that our technical contribution is limited since Theorem 3.1 is straightforward given Proposition 2.2 presented in [44, Corollary 4.2]. We emphasize that our technical contribution is not only Theorem 3.1 but, more importantly, developing fast algorithms with predictions for Laminar, Nested, and Box in Section 4 using Theorem 3.1.
Specifically, to obtain our $\mathrm{O}(n \\|x^* - \hat x\\|_1)$-time algorithm for Laminar, we have developed

- an $\mathrm{O}(n)$-time projection method by carefully modifying the $\mathrm{O}(n\log^2 n)$-time convex min-sum convolution [45] (see Section 4.1) and
- an $\mathrm{O}(n)$-time dynamic programming method for computing the steepest descent direction by refining the method in [30] (see Section 4.2). (The direct application of [30] costs $\mathrm{O}(n^2)$ time.)

As for Box, we achieved faster $\mathrm{O}(\log n)$-time direction finding in Section 4.3 by using the min-heap technique. While the technique is standard, this improvement is crucial for our method for Box to enjoy the potential to go beyond the lower-bound result [19] with moderately accurate predictions, as discussed in Section 1.1. Our polynomial-time projection and direction finding for general M-convex minimization in Section 3.1 and Appendix A.1 also involve substantial technical sophistication.

All in all, despite the existence of (Sakaue and Oki, 2022) and other existing studies [30, 44, 45], our work has significant technical novelty, which is crucial for the potential of our time complexity bounds in Table 1 to surpass the best worst-case and lower-bound results.

---

### Decision · Program_Chairs · 2023-09-21

**Decision:**

Accept (poster)

**Comment:**

Summary and contributions:
======================
This paper studies an acceleration for M-convex function minimization extending recent findings from Sakaue and Oki 2022 and warm starting L-convex function minimization with predictions.

The proposed methodology enhances pre-existing worst case time complexity and where the time-complexity bound are a function of the L1 distance between the predicted initial solution and the unique optimal solution. The authors introduced faster algorithms for laminar, nested, and box settings.

Review Summary:
=============
Two of the initial reviewers gave low confidence reviews. Alternative emergency recruiters with sufficient experience were recruited awarding the paper an accept score (7) with high confidence (4). The remaining initial reviewers awarded the paper 5/6 (borderline/weak accept) with confidence of 3.

There is a reasonable consensus among the reviewers with common strengths being observed where the contributions (above) were identified.

Common weaknesses included:

* CW1: A suggestion that the technical contribution is limited (i.e. an adaptation of S+O 2022)
* CW2: Lacking sufficient empirical results. Including testing on standard test sets.
* CW3: An observation that the worst case complexity depends on the accuracy of the initial predictions as the time is bounded by the L1 distance of initial and optimal solutions.

Responses from the authors addressed these concerns resulting in the reviewers increasing their scores.

* CW1: The authors rebuttal stated that their technical contributions is not only limited to the solution distance complexity (Thm3.1), but also algorithm speedups for O(n) time projection method for convex min-sum convolution (S4.1) and O(n) DP method for computing steepest descent direction (S4.2).

* CW2: Authors provided additional experimental results for the nested problem from Wu et al.[47] with F, FUEL and CRASH objective functions during the rebuttal phase.

* CW3: I believe this observation to be a non-issue. The corollary of this is that with very accurate predictions, the performance may exceed lower bound of classical optimization algorithms.


Reason for decision:
=================
Consistent reviews 5,6,7,7 with confidence 3,3,4,4. Most weaknesses addressed resulted in reviewers increasing scores.